# Modulation of dopamine D₁ receptors via histamine H₃ receptors is a novel therapeutic target for Huntington's disease

David Moreno-Delgado[1,2†‡], Mar Puigdellívol[2,3,4,5†§], Estefanía Moreno[1,2], Mar Rodríguez-Ruiz[1,2], Joaquín Botta[1,5], Paola Gasperini[5], Anna Chiarlone[2,6], Lesley A Howell[7], Marco Scarselli[8], Vicent Casadó[1,2], Antoni Cortés[1,2], Sergi Ferré[9], Manuel Guzmán[2,6], Carmen Lluís[1,2], Jordi Alberch[2,3,4], Enric I Canela[1,2], Silvia Ginés[2,3,4†*], Peter J McCormick[1,2,5,10†*]

[1]Department of Biochemistry and Molecular Biomedicine, Faculty of Biology, Institute of Biomedicine of the University of Barcelona (IBUB), University of Barcelona, Barcelona, Spain; [2]Centro de Investigación Biomédica en Red sobre Enfermedades Neurodegenerativas, Madrid, Spain; [3]Department of Biomedical Science, Faculty of Medicine, University of Barcelona, Institut of Neuroscience, Barcelona, Spain; [4]Institut d´Investigacions Biomèdiques August Pi i Sunyer (IDIBAPS), Barcelona, Spain; [5]School of Pharmacy, University of East Anglia, Norwich Research Park, Norwich, United Kingdom; [6]Department of Biochemistry and Molecular Biology I, School of Biology, Instituto Universitario de Investigación Neuroquímica, and Instituto Ramón y Cajal de Investigación Sanitaria, Complutense University of Madrid, Madrid, Spain; [7]School of Biological and Chemical Sciences, Queen Mary University of London, London, United Kingdom; [8]Department of Translational Research and New Technologies in Medicine and Surgery, University of Pisa, Pisa, Italy; [9]National Institute on Drug Abuse, Intramural Research Program, National Institutes of Health, Department of Health and Human Services, Baltimore, United States; [10]William Harvey Research Institute, Barts and the London School of Medicine, Queen Mary University of London, London, United Kingdom

*For correspondence:
silviagines@ub.edu (SG);
p.mccormick@qmul.ac.uk (PJMC)

†These authors contributed equally to this work

Present address: ‡UCB BioPharma SPRL, Chemin de Foriest, Braine-l'Alleud, Braine-l'Alleud, Belgium; §Department of Biochemistry, University of Cambridge, Cambridge, United Kingdom

Competing interests: The authors declare that no competing interests exist.

**Abstract** Early Huntington's disease (HD) include over-activation of dopamine D₁ receptors (D₁R), producing an imbalance in dopaminergic neurotransmission and cell death. To reduce D₁R over-activation, we present a strategy based on targeting complexes of D₁R and histamine H₃ receptors (H₃R). Using an HD mouse striatal cell model and HD mouse organotypic brain slices we found that D₁R-induced cell death signaling and neuronal degeneration, are mitigated by an H₃R antagonist. We demonstrate that the D₁R-H₃R heteromer is expressed in HD mice at early but not late stages of HD, correlating with HD progression. In accordance, we found this target expressed in human control subjects and low-grade HD patients. Finally, treatment of HD mice with an H₃R antagonist prevented cognitive and motor learning deficits and the loss of heteromer expression. Taken together, our results indicate that D₁R - H₃R heteromers play a pivotal role in dopamine signaling and represent novel targets for treating HD.

## Introduction

Huntington's disease (HD) is a dominant inherited progressive neurodegenerative disorder caused by expansion of a CAG repeat, coding a polyglutamine repeat within the *N*-terminal region of huntingtin protein (*Macdonald, 1993*; *Vonsattel and DiFiglia, 1998*). Although dysfunction and death of striatal medium-sized spiny neurons (MSSNs) is a key neuropathological hallmark of HD (*Ferrante et al., 1991*; *Vonsattel et al., 1985*), cognitive deficits appear long before the onset of motor disturbances (*Lawrence et al., 2000*; *Lemiere et al., 2004*). It has been postulated that alterations in the dopaminergic system may contribute to HD neuropathology (*Chen et al., 2013a*; *Jakel and Maragos, 2000*), as dopamine (DA) plays a key role in the control of coordinated movements. Increased DA levels and DA signaling occur at early stages of the disease (*Chen et al., 2013a*; *Garret et al., 1992*; *Jakel and Maragos, 2000*), resulting in an imbalance in striatal neurotransmission initiating signaling cascades that may contribute to striatal cell death (*Paoletti et al., 2008*; *Ross and Tabrizi, 2011*). Several studies demonstrated that DA receptor antagonists and agents that decrease DA content reduce chorea and motor symptoms while dopaminergic stimulation exacerbate such symptoms (*Huntington Study Group, 2006*; *Mestre et al., 2009*; *Tang et al., 2007*).

Within the striatum, two different MSSNs populations can be distinguished: 1) MSSNs expressing enkephalin and dopamine $D_2$ receptors ($D_2R$), which give rise to the indirect striatal efferent pathway, and 2) MSSNs expressing substance P and dopamine $D_1$ receptors ($D_1R$), comprising the direct striatal efferent pathway. Recently, several studies with experimental models have changed the traditional view that $D_2R$-MSSNs are more vulnerable in HD (*Cepeda et al., 2008*; *Kreitzer and Malenka, 2007*), proposing a new view in which $D_1R$-MSSNs are more vulnerable to the HD mutation. In this view, it has been demonstrated that mutant huntingtin enhances striatal cell death through the activation of $D_1R$ but not $D_2R$ (*Paoletti et al., 2008*). More recently, it has been described that, at early stages of the disease, HD mice show an increase in glutamate release onto $D_1R$ neurons but not $D_2R$ neurons while, later in the disease, glutamate release is selectively decreased to $D_1R$ cells (*André et al., 2011a*), indicating that several changes occur in $D_1R$ neurons at both early and late disease stages. Strategies that might reduce $D_1R$ signaling could prove successful towards preventing HD (*André et al., 2011a*; *André et al., 2011b*; *Ross and Tabrizi, 2011*; *Tang et al., 2007*). However, $D_1Rs$ are highly expressed in many tissues (*Beaulieu and Gainetdinov, 2011*) and broad use of $D_1R$ antagonists as a preventive treatment has important drawbacks including locomotor impairments (*Giménez-Llort et al., 1997*), or induce depression, parkinsonism and sedation in HD patients (*Frank et al., 2008*; *Huntington Study Group, 2006*).

Histamine is an important neuromodulator with four known G protein-coupled receptors (GPCRs). $H_3Rs$ are expressed in brain regions involved in both motor function (striatum) and cognition, such as the cortex, thalamus, hypothalamus, hippocampus and amygdala (*Panula and Nuutinen, 2013*). It is known that in at least striatal GABAergic dynorphinergic neurons (*Pillot et al., 2002*; *Ryu et al., 1994a*; *Ryu et al., 1994b*), both $D_1R$ and $H_3R$ are co-expressed and we and others have found that they establish functional negative interactions by forming molecular complexes termed heteromers (*Moreno et al., 2011*; *Sánchez-Lemus and Arias-Montaño, 2004*). Hence, in this work, we hypothesized that targeting $D_1R$ through these receptor complexes of $D_1R$ and $H_3R$ might serve as a more efficient and targeted strategy to slow the progression of HD. Specifically, we demonstrate that $D_1R$-$H_3R$ heteromers are expressed and functional in early HD stages but are lost in late stages. An $H_3R$ antagonist acting through $D_1R$-$H_3R$ heteromers acts as a protective agent against dopaminergic imbalance in early HD stages improving learning and long-term memory deficits and rescuing the loss of $D_1R$-$H_3R$ complexes at late stages of HD.

## Results

### Functional $D_1R$-$H_3R$ heteromers are expressed in wild type STHdh$^{Q7}$ and HD STHdh$^{Q111}$ striatal cell models

To test whether $D_1R$-$H_3R$ heteromers could indeed be targets for controlling $D_1R$ signaling in HD, we first analyzed the expression of both receptors in immortalized striatal cells expressing endogenous levels of full-length wild-type STHdh$^{Q7}$ or mutant STHdh$^{Q111}$ huntingtin (*Ginés et al., 2010*). Ligand binding determined that both STHdh$^{Q7}$ and STHdh$^{Q111}$ cells endogenously express similar

levels of $D_1R$ and $H_3R$ (*Supplementary file 1*). By proximity ligation assays (PLA), $D_1R$-$H_3R$ heteromers were detected as red spots surrounding the blue stained nuclei in both cell types (*Figure 1A*, left panels of both cell types) and in cells treated with control lentivirus vector (*Figure 1—figure supplement 1A*) but not in cells depleted of $H_3R$ (*Figure 1A*, right panels of both cell types) by shRNA, as shown by RT-PCR and functionality (*Figure 1—figure supplement 1B,C*), or in negative controls (*Figure 1—figure supplement 1D*). To ensure that $D_1R$-$H_3R$ heteromers were functional in STHdh cells, cell signaling experiments were performed. Using both STHdh$^{Q7}$ and STHdh$^{Q111}$ cells and concentrations of ligands previously shown to be optimal for receptor activation of the ERK1/2 pathway (*Ferrada et al., 2009*; *Moreno et al., 2014*; *Moreno et al., 2011*), we observed that the $D_1R$ agonist SKF 81297 was able to increase ERK1/2 phosphorylation whereas it was prevented by $D_1R$ antagonist SCH 23390, and by the $H_3R$ antagonist thioperamide (*Figure 1—figure supplement 2A, B*) via cross-antagonism. In addition, we tested a previously described alternative signaling pathway activated downstream of $D_1R$, $Ca^{2+}$ mobilization (*Chen et al., 2007*; *Jose et al., 1995*). When cells were treated with the $D_1R$ agonist SKF 81297 a robust and rapid increase in cytosolic $Ca^{2+}$ was detected in both STHdh$^{Q7}$ and STHdh$^{Q111}$ cells (*Figure 1B,C*). Importantly, this calcium release could be dampened with the $H_3R$ antagonist thioperamide (cross-antagonism) (*Figure 1B,C*). The above signaling data strongly support the presence of functional $D_1R$-$H_3R$ heteromers in STHdh cells.

To further demonstrate that an $H_3R$ antagonist is dampening $D_1R$ activation involving $D_1R$-$H_3R$ heteromers, we evaluated the effect of interfering peptides, which are synthetic peptides with the amino acid sequence of domains of the receptors involved in the heteromeric interface. This approach has been used by us and others to disrupt other heteromer complexes (*Bonaventura et al., 2015*; *Guitart et al., 2014*; *Hasbi et al., 2014*; *Lee et al., 2014*; *Viñals et al., 2015*). In a previous study we showed the efficacy of this approach in demonstrating heteromerization of $D_1R$ with $D_3R$, using a peptide with the sequence of $D_1R$ transmembrane domain 5 (TM5) but not TM7 (*Guitart et al., 2014*). We therefore investigated whether synthetic peptides with the sequence of TM5, and TM7 (as a negative control) of $D_1R$, fused to HIV-TAT, were also able to disrupt receptor $D_1R$-$H_3R$ heteromers measured by PLA. In agreement with our hypothesis, there was a near complete loss in PLA fluorescence signal when STHdh$^{Q7}$ and STHdh$^{Q111}$ cells were incubated with TAT-TM five peptide (*Figure 1D,F*), but not for the negative control in which the TAT-TM seven peptide was used (*Figure 1H,J*). We next evaluated whether TM5 or TM7 would interfere with the observed cross-antagonism in calcium mobilization assays. Clearly, pretreatment of both STHdh$^{Q7}$ and STHdh$^{Q111}$ cells with the TAT-TM5 (*Figure 1E,G*) but not TAT-TM7 (*Figure 1I,K*) peptide disrupts the ability of the $H_3R$ antagonist thioperamide to dampen $D_1R$ calcium signaling. These results support that TM5 forms part of the interface of the $D_1R$-$H_3R$ heteromer and demonstrate that the $H_3R$ antagonist effect is driven through direct interaction between $D_1R$ and $H_3R$.

## $H_3R$ ligands prevent the $D_1R$-induced cell death in STHdh$^{Q7}$ and STHd$^{Q111}$ cells

It has been previously reported that upon activation of $D_1R$, STHdh cell viability is reduced (*Paoletti et al., 2008*). To explore whether $H_3R$ ligands could impair $D_1R$ activation through $D_1R$-$H_3R$ heteromers in a pathologically relevant readout, we used $D_1R$-induced cell death as an output of $D_1R$ activation in STHdh cells. As expected, STHdh cell viability decreased when treated with the $D_1R$ agonist SKF 81297 in a concentration-dependent manner (*Figure 1—figure supplement 2C*). Significant cell death did not occur until 30 µM SKF 81297 was used (*Figure 1—figure supplement 2C*), an effect prevented by the $D_1R$ antagonist SCH 23390 (*Figure 1—figure supplement 2E*). Pretreatment with the $H_3R$ antagonist thioperamide, which did not modify cell viability when administered alone (*Figure 1—figure supplement 2E*), increased the number of surviving cells in the presence of the $D_1R$ agonist SKF 81297 in both cell types (*Figure 1L,M* and *Figure 1—figure supplement 2D*). Importantly, the effect of the $H_3R$ antagonist thioperamide was specific since no protection from $D_1R$ agonist-induced cell death was observed in cells depleted of $H_3R$ with shRNA lentiviral infection (*Figure 1L,M*), but was observed in cells transfected with the control lentivirus (*Figure 1—figure supplement 2F*). In addition, we also demonstrated that recovery of viability induced by the $H_3R$ antagonist thioperamide was mediated by $D_1R$-$H_3R$ heteromers since pre-incubation with $D_1R$ TM5 peptide, but not $D_1R$ TM7 impaired the $H_3R$ antagonist protection from $D_1R$ agonist-induced cell death (*Figure 1L,M*).

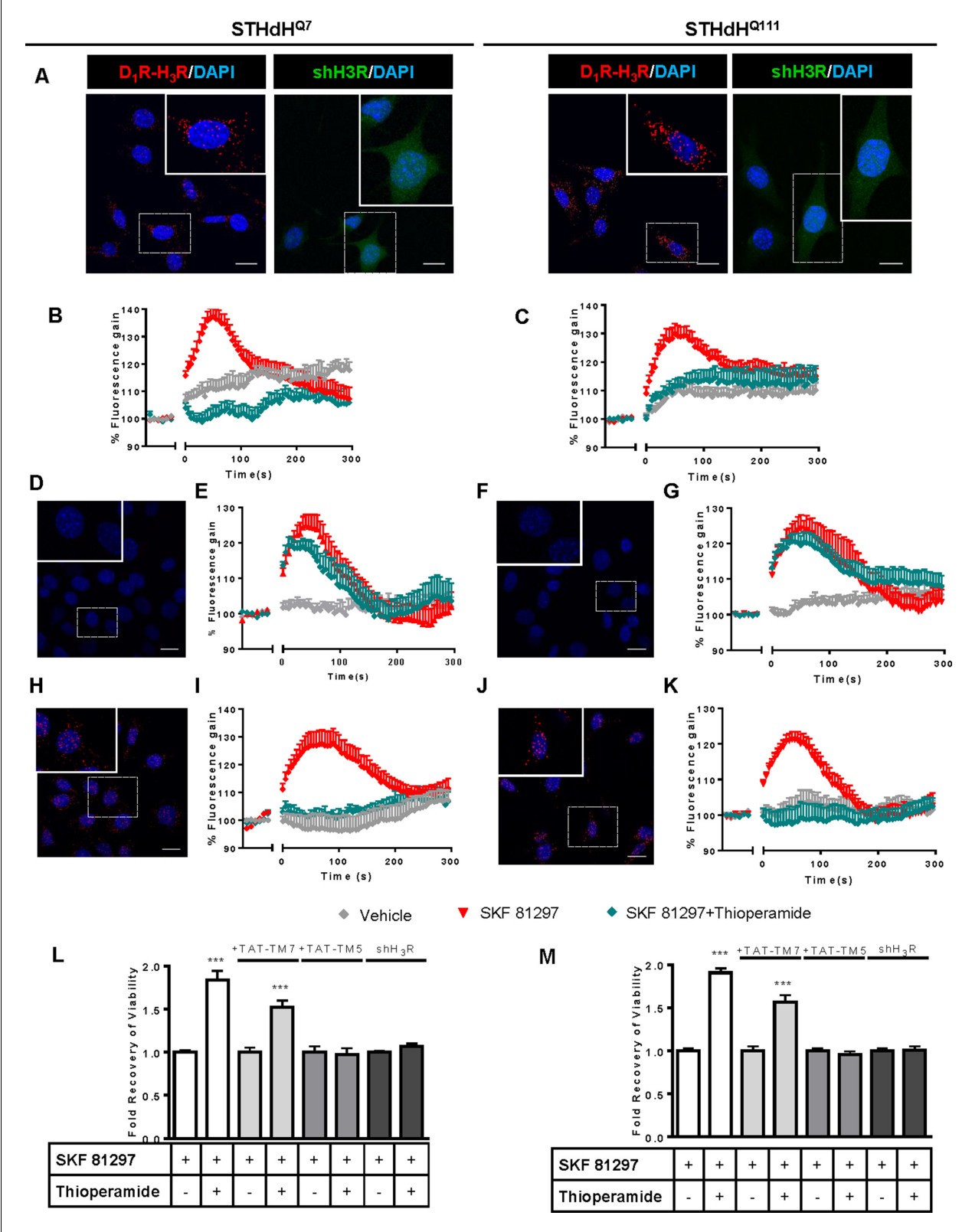

**Figure 1.** Functional $D_1R$-$H_3R$ heteromers are expressed in STHdh$^{Q7}$ and STHdh$^{Q111}$ cells. PLA were performed in STHdh$^{Q7}$ and STHdh$^{Q111}$ cells (A, D, F, H and J) or in cells infected with shH$_3$R to silence H$_3$R, observed as green stained cells due to the GFP expression included in the plasmid (A). $D_1R$-H$_3$R heteromers were visualized in STHdh cells as red spots around blue colored DAPI stained nucleus, but not in STHdh cells infected with shH$_3$R vector (A). Calcium increases were measured in STHdh$^{Q7}$ (B, E and I) or STHdh$^{Q111}$ (C, G and K). Cells were treated (20 min) or not with the H$_3$R

*Figure 1 continued on next page*

*Figure 1 continued*

antagonist thioperamide (10 μM) before the addition of vehicle or SKF 81297 (1 μM). In (D, E, F, G, H, I, J and K), STHdH$^{Q7}$ (D, E, H and I) or STHdH$^{Q111}$ (F, G, J and K) cells were also pre-treated for 60 min with 4 μM TM5 (D, E, F and G) or TM7 (H, I, J and K) peptides. Heteromers were visualized as red spots around DAPI (blue) stained nucleus in cells pre-treated with TM7 peptide. Scale: 20 μm. For each calcium curve values are expressed as a percentage increase with respect to untreated cells and are a mean ± SEM of 3 to 5 independent experiments. In (L and M), cell viability was determined in STHdh$^{Q7}$ (L) or STHdh$^{Q111}$ cells (M) pre-treated for 60 min with vehicle (white columns), with 4 μM TAT-TM7 (pale grey columns) or TAT-TM5 (grey columns) or infected with shH$_3$R to silence H$_3$R (dark grey columns) prior overstimulation with 30 μM SKF 81297. Values represent mean ± SEM (n = 24 to 30) of cell viability recovery expressed as in-fold respect to SKF 81297 treated cells. Student's *t* test showed a significant (***p<0.001) effect over SKF 81297 treated cells.

The online version of this article includes the following figure supplement(s) for figure 1:

**Figure supplement 1.** Negative controls for Proximity Ligation Assays (PLA) in striatal cells not depleted or H$_3$R depleted by shRNA.

**Figure supplement 2.** H$_3$R ligands revert the D$_1$R-mediated decreases in STHdh$^{Q7}$ and STHdh$^{Q111}$ cell viability.

**Figure supplement 3.** Effect of low and high SKF 81297 concentrations in p-p38 and intracellular calcium release.

**Figure supplement 4.** H$_3$R ligands revert the D$_1$R-mediated decreases in cell viability in STHdh$^{Q7}$ and STHdh$^{Q111}$ by modulating calcium signaling and p38 phosphorylation.

**Figure supplement 5.** H$_3$R ligands revert the D$_1$R overstimulation-induced heteromer disruption in striatal cells.

To better understand the mechanisms involved in D$_1$R-H$_3$R heteromer action, we determined which cellular signaling pathways are implicated in the cross-antagonism of H$_3$R upon activation of D$_1$R. Both concentrations of the D$_1$R agonist SKF 81297, cytotoxic (30 μM) and non-cytotoxic (1 μM), can induce intracellular calcium release, which is more pronounced and persistent at 30 μM (*Figure 1—figure supplement 3A,B*). A correlation between the intensity of calcium responses and the activation of apoptotic pathways such as p38 (*Semenova et al., 2007*) has been previously demonstrated. Thus, we measured changes in p38 phosphorylation levels using both concentrations of the D$_1$R agonist SKF 81297 (*Figure 1—figure supplement 3C,D*). Interestingly, we found that increased phosphorylation of p38 only occurred at the cytotoxic concentration of SKF 81297. Similar to treatment with 1 μM SKF 81297 (see *Figure 1*), the calcium release induced by 30 μM SKF 81297 was also blocked by the H$_3$R antagonist thioperamide (*Figure 1—figure supplement 4A,B*). Treatment with the H$_3$R antagonist thioperamide reduced p38 phosphorylation upon D$_1$R activation in both cell types (*Figure 1—figure supplement 4C*). Moreover, the p38 inhibitor SB 203580 blocked p38 phosphorylation (*Figure 1—figure supplement 4C*) and protected against the cytotoxic effect of the D$_1$R agonist SKF 81297 in a dose-dependent manner (*Figure 1—figure supplement 4D*), confirming that p38 is a key pathway involved in D$_1$R-mediated cell death in these cells.

It has been reported that ligands can influence receptor oligomerization. To understand how the ligands used here might impact D$_1$R-H$_3$R heteromers we performed PLA after treating with either vehicle, SKF 81297 or SKF 81297 and thioperamide. We found that SKF 81297-induced a loss of PLA staining in both STHdh cells (*Figure 1—figure supplement 5*), while pre-treatment with the H$_3$R antagonist thioperamide preserved the number of punctate PLA spots (*Figure 1—figure supplement 5*).

## Functional D$_1$R-H$_3$R heteromers are expressed in wild-type Hdh$^{Q7/Q7}$ and in Hdh$^{Q7/Q111}$ mutant knock-in mice at early but not late HD stages

To test whether D$_1$R-H$_3$R heteromers can indeed be targets for treating HD, we investigated their expression and function in the striatum, cerebral cortex and hippocampus of a widely accepted pre-clinical model of HD, the heterozygous Hdh$^{Q7/Q111}$ mutant knock-in mice, and their wild-type Hdh$^{Q7/Q7}$ littermates (*Giralt et al., 2012*; *Puigdellívol et al., 2015*). By PLA we confirmed that both Hdh$^{Q7/Q7}$ and Hdh$^{Q7/Q111}$ mice display D$_1$R-H$_3$R heteromers at 2 months (mo) (*Figure 2—figure supplement 1*) and four mo (*Figure 2A,B*) of age in all brain regions tested. No signal was observed in negative controls in which one of the PLA primary antibodies were missing (*Figure 2—figure supplement 2*). Heteromer expression was similar in all brain areas and no differences were observed between genotypes at 4 mo of age (*Figure 2B*). Surprisingly, an almost complete loss of D$_1$R-H$_3$R heteromers was found in 6 mo and eight mo-old Hdh$^{Q7/Q111}$ mice but not in Hdh$^{Q7/Q7}$ mice

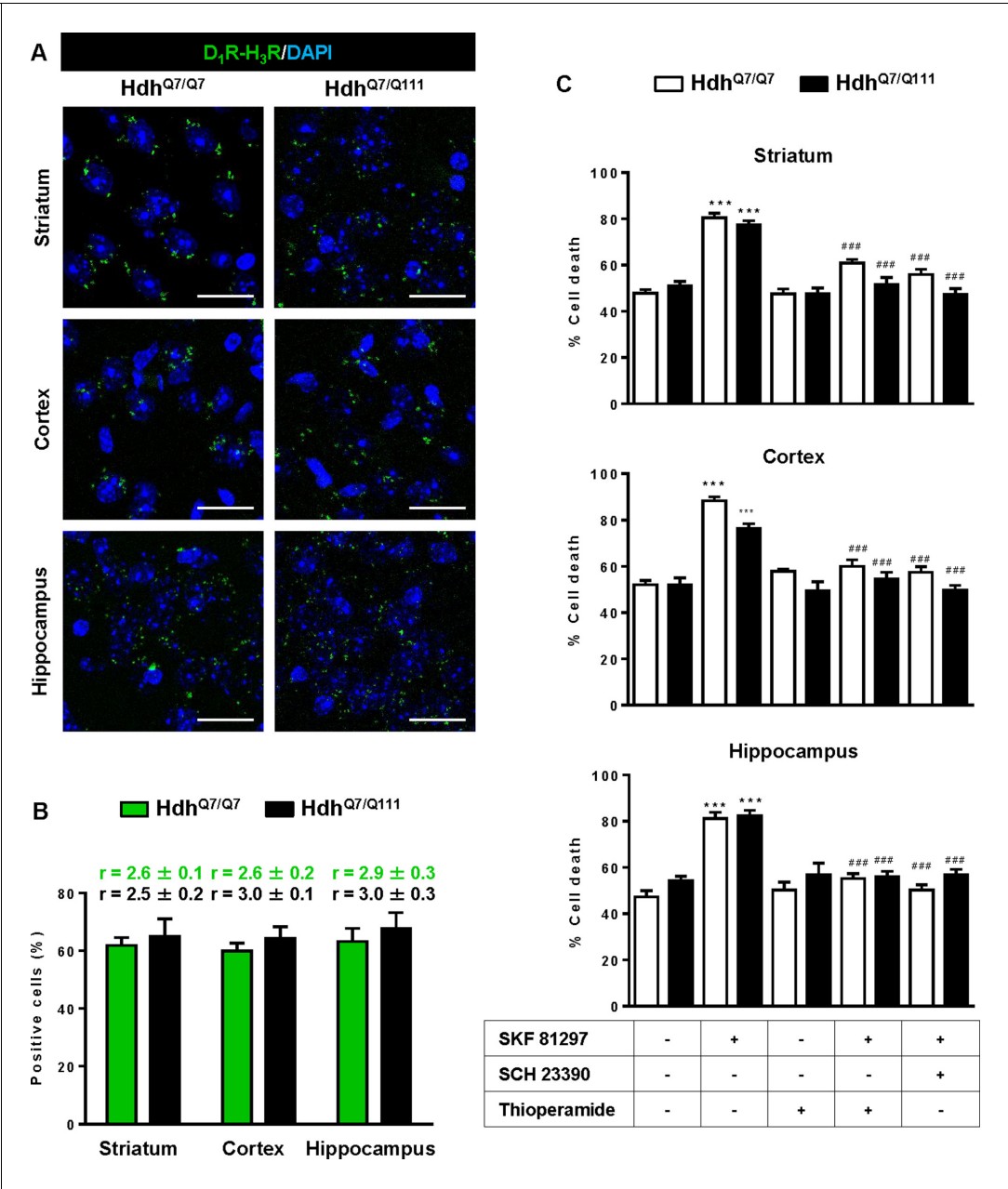

**Figure 2.** Functional $D_1R$-$H_3R$ heteromers are expressed in wild-type $Hdh^{Q7/Q7}$ and mutant $Hdh^{Q7/Q111}$ mice. Striatal, cortical or hippocampal slices from 4-month-old $Hdh^{Q7/Q7}$ and $Hdh^{Q7/Q111}$ mice were used. In (**A**), by Proximity Ligation Assays (PLA) $D_1R$-$H_3R$ heteromers were visualized in all slices as green spots around blue colored DAPI stained nucleus. Scale bar: 20 μm. In (**B**), the number of cells containing one or more green spots is expressed as the percentage of the total number of cells (blue nucleus). *r values* (number of green spots/cell containing spots) are shown above each bar. Data (% of positive cells or r) are the mean ± SEM of counts in 600–800 cells from 4 to 8 different fields from three different animals. Student's *t* test showed no significant differences in heteromers expression in $Hdh^{Q7/Q7}$ and $Hdh^{Q7/Q111}$ mice. In (**C**), striatal, cortical or hippocampal organotypic slice cultures from 4-month-old $Hdh^{Q7/Q7}$ and $Hdh^{Q7/Q111}$ mice were treated for 60 min with vehicle, the $D_1R$ antagonist SCH 23390 (10 μM) or $H_3R$ antagonist thioperamide (10 μM) before the addition of SKF 81297 (50 μM). After 48 h cell death was determined. Values represent mean ± SEM (n = 3 to 19) of percentage of cell death. One-way ANOVA followed by Bonferroni *post hoc* tests showed a significant effect over non-treated organotypic cultures (\*\*\*p<0.001) or of the $H_3R$ antagonist plus SKF 81297 treatment over the SKF 81297 (###p<0.001).

The online version of this article includes the following figure supplement(s) for figure 2:

**Figure supplement 1.** $D_1R$-$H_3R$ heteromer are expressed in 2-month-old $Hdh^{Q7/Q7}$ and $Hdh^{Q7/Q111}$ mice.

**Figure supplement 2.** Negative controls for Proximity Ligation Assays (PLA) in mouse brain slices.

(*Figure 3—figure supplement 1* and *Figure 3A,B*), indicating that at more advanced disease stages the $D_1R$-$H_3R$ heteromer is lost. Although at 8 mo of age we detected a partial decrease in striatal $D_1R$ expression in $Hdh^{Q7/Q111}$ compared with $Hdh^{Q7/Q7}$ mice using ligand binding experiments (*Supplementary file 2*), the loss of heteromer expression is not due to a complete loss of receptor expression since by radioligand binding (*Supplementary file 2*) and mRNA expression analysis (*Supplementary file 3*) both receptors continue to be expressed.

To test the role of $D_1R$-$H_3R$ heteromers, organotypic mouse striatal, cortical and hippocampal cultures were obtained. Cell death was induced by the $D_1R$ agonist SKF 81297 (50 µM), and analysis of DAPI and propidium iodide staining was performed. As expected, $D_1R$ agonist SKF 81297 treatment increased the percentage of cell death in all three regions compared to vehicle-treated organotypic cultures without significant differences between genotypes at 4 mo of age (*Figure 2C*). Importantly, slices pre-treated with the $H_3R$ antagonist thioperamide, that does not modify cell death when administered alone, protected cells from $D_1R$ elicited cell death in an equivalent manner to the $D_1R$ antagonist SCH 23390 (*Figure 2C*), indicating that functional $D_1R$-$H_3R$ heteromers are expressed in different brain areas of $Hdh^{Q7/Q7}$ and $Hdh^{Q7/Q111}$ mice at early disease stages. The dramatic change in heteromer expression in eight mo-old $Hdh^{Q7/Q111}$ mice was mirrored by the lack of protection of the $H_3R$ antagonist thioperamide against SKF 81297-induced cell death in organotypic cultures (*Figure 3C*), corroborating that the presence of $D_1R$-$H_3R$ heteromers is needed for the $H_3R$ antagonist to prevent $D_1R$-mediated cell death.

## Treatment with thioperamide prevents cognitive and motor learning deficits at early disease stages

To test whether the $H_3R$ antagonist thioperamide can exert beneficial effects in the initial stages of the disease we evaluated the effect of chronic thioperamide treatment on motor learning and memory deficits in mutant $Hdh^{Q7/Q111}$ mice. Since cognitive decline is observed in these HD mice from 6 mo of age (*Brito et al., 2014*; *Giralt et al., 2012*; *Puigdellívol et al., 2015*) and the $D_1R$-$H_3R$ heteromers are expressed and functional until the age of 5 mo (*Figure 4—figure supplement 1A,B*), we chose 5mo-old animals to start the thioperamide treatment (*Figure 4—figure supplement 2*). Corticostriatal function in saline and thioperamide-treated $Hdh^{Q7/Q7}$ and $Hdh^{Q7/Q111}$ mice was analyzed by using the accelerating rotarod task that evaluates the acquisition of new motor skills (*Puigdellívol et al., 2015*). Saline-treated mutant $Hdh^{Q7/Q111}$ mice were unable to maintain their balance on the rotarod as wild-type $Hdh^{Q7/Q7}$ mice revealing impaired acquisition of new motor skills (*Figure 4A*). Chronic treatment with thioperamide completely rescued motor learning deficits in mutant $Hdh^{Q7/Q111}$ mice as evidenced by a similar latency to fall in the accelerating rotarod as wild-type $Hdh^{Q7/Q7}$ mice. Next, recognition long-term memory (LTM) was analyzed by using the novel object recognition test (NORT) (*Figure 4B*). After two days of habituation in the open field arena (*Figure 4—figure supplement 3*), no significant differences were found between genotypes and/or treatments, demonstrating no alterations in motivation, anxiety or spontaneous locomotor activity. After habituation, animals were subjected to a training session in the open field arena in the presence of two similar objects (A and A'). Both saline and thioperamide-treated wild-type $Hdh^{Q7/Q7}$ and mutant $Hdh^{Q7/Q111}$ mice similarly explored both objects indicating neither object nor place preferences (*Figure 4B*). After 24 hr, LTM was evaluated by changing one of the old objects (A') for a novel one (B). Whereas saline-treated $Hdh^{Q7/Q111}$ mice did not show any preference for the novel object with respect to the familiar one, indicating recognition LTM deficits, thioperamide treatment completely prevented this LTM deficit in mutant $Hdh^{Q7/Q111}$ mice (*Figure 4B*). Next, spatial LTM was analyzed using the T-maze spontaneous alternation task (T-SAT) (*Figure 4C*). During the training, similar exploration time (*Figure 4C*, left panel) and similar number of arm entries (*Figure 4—figure supplement 4*, *left panel*) were found in all genotypes and treatments. After 5 hr, a testing session showed that saline-treated $Hdh^{Q7/Q111}$ mice had no preferences between the novel arm and the old arm, indicating spatial LTM deficits (*Figure 4C*, right panel). Interestingly, mutant $Hdh^{Q7/Q111}$ mice treated with thioperamide spent more time in the novel *versus* the old arm, revealing preserved LTM (*Figure 4C*, right panel). Overall, these data demonstrate the effectiveness of thioperamide treatment in restoring motor learning and preventing spatial and recognition LTM deficits in mutant $Hdh^{Q7/Q111}$ mice.

We next tested if the reversion of the HD phenotype in mutant $Hdh^{Q7/Q111}$ mice induced by thioperamide treatment correlated with the preservation of $D_1R$-$H_3R$ heteromer expression. By PLA we

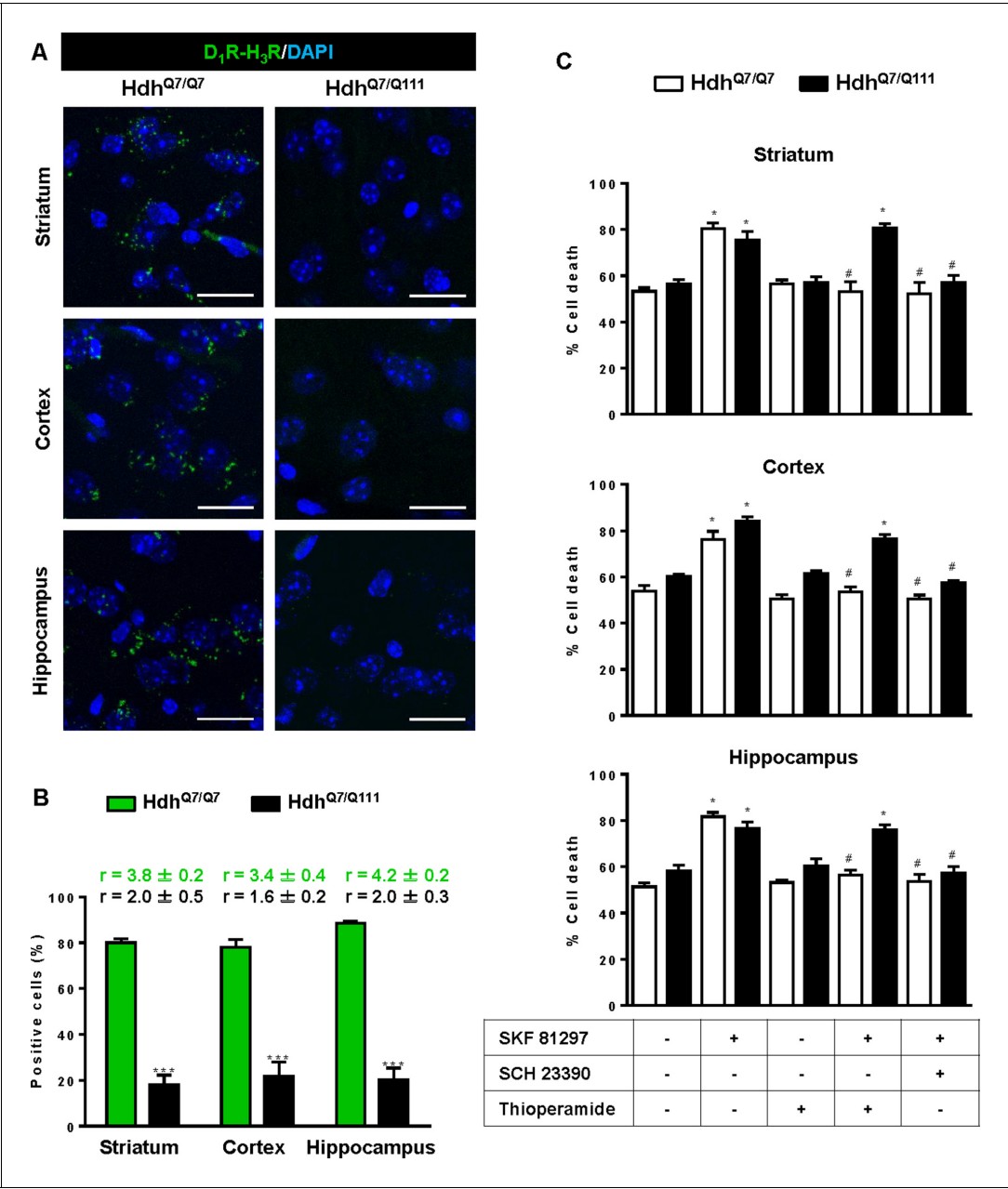

**Figure 3.** Functional $D_1R$-$H_3R$ heteromers are expressed in wild-type $Hdh^{Q7/Q7}$ but not in 8-month-old mutant $Hdh^{Q7/Q111}$ mice. Striatal, cortical or hippocampal slices from 8-month-old $Hdh^{Q7/Q7}$ and $Hdh^{Q7/Q111}$ mice were used. In (A), by Proximity Ligation Assays (PLA) $D_1R$-$H_3R$ heteromers were visualized in $Hdh^{Q7/Q7}$ mice but not in $Hdh^{Q7/Q111}$ mice as green spots around blue colored DAPI stained nucleus. Scale bar: 20 $\mu$m. In (B), the number of cells containing one or more green spots is expressed as the percentage of the total number of cells (blue nucleus). *r values* (number of green spots/ cell containing spots) are shown above each bar. Data (% of positive cells or r) are the mean ± SEM of counts in 600–800 cells from 5 to 7 different fields from three different animals. Student's *t* test showed a significant (\*\*\*p<0.05) decrease of heteromers expression in $Hdh^{Q7/Q111}$ mice compared to the respective $Hdh^{Q7/Q7}$ mice. In (C) striatal, cortical or hippocampal organotypic slice cultures from 8-month-old $Hdh^{Q7/Q7}$ and $Hdh^{Q7/Q111}$ mice were treated for 60 min with medium, the $D_1R$ antagonist SCH 23390 (10 $\mu$M) or the $H_3R$ antagonist thioperamide (10 $\mu$M) before the addition of SKF 81297 (50 $\mu$M) and cell death was determined. Values represent mean ± SEM (n = 3 to 6) of percentage of cell death. One-way ANOVA followed by Bonferroni *post hoc* tests showed a significant effect over non-treated organotypic cultures (\*p<0.05) or of the $H_3R$ antagonist plus SKF 81297 treatment over the SKF 81297 (#p<0.05).

The online version of this article includes the following figure supplement(s) for figure 3:

**Figure supplement 1.** Expression of $D_1R$-$H_3R$ heteromers in 6-month-old $Hdh^{Q7/Q7}$ and $Hdh^{Q7/Q111}$ mice chronically treated with saline.

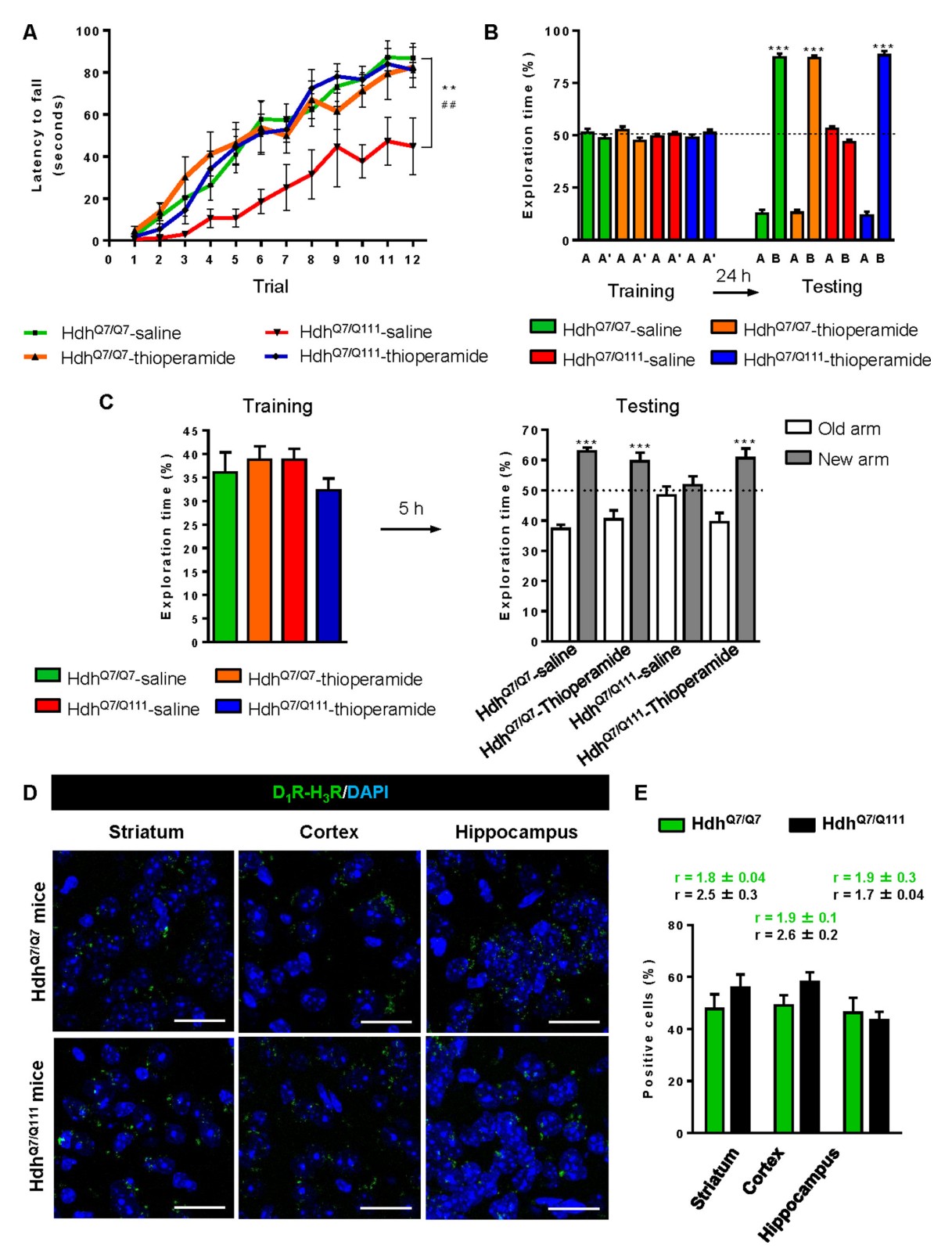

**Figure 4.** Thioperamide chronic treatment prevents motor learning, long-term memory (LTM) deficits and the loss of receptor heteromerization in 6-month-old Hdh$^{Q7/Q111}$ mice. In (**A**), curves illustrating the latency to fall in the accelerating rotarod of 6-month-old Hdh$^{Q7/Q7}$ and Hdh$^{Q7/Q111}$ mice treated with saline or thioperamide from 5 months of age are shown. In (**B**), the exploration time for saline or thioperamide-treated Hdh$^{Q7/Q7}$ and Hdh$^{Q7/Q111}$ mice during the training and the testing (24 hr delay, LTM) sessions in a novel-object recognition task showing that long-term recognition

*Figure 4 continued on next page*

*Figure 4 continued*

memory deficits are rescued in the thioperamide-treated Hdh$^{Q7/Q111}$ mice. One-way ANOVA with Bonferroni *post hoc* showed significant differences (\*\*\*p<0.001) compared to the old object recognition. In (C), bar diagram illustrating the exploration time for saline- or thioperamide-treated Hdh$^{Q7/Q7}$ and Hdh$^{Q7/Q111}$ mice during the training and the 5 hr later testing in the T-SAT showing thioperamide reverses spatial long-term memory (LTM) deficits. In (A) to C), 11 saline-treated Hdh$^{Q7/Q7}$ mice, 10 thioperamide-treated Hdh$^{Q7/Q7}$ mice, seven saline-treated Hdh$^{Q7/Q111}$ mice and nine thioperamide-treated Hdh$^{Q7/Q111}$ mice were evaluated at 6 months of age. In (D) PLA were performed in striatal, cortical and hippocampal slices from 6-month-old Hdh$^{Q7/Q7}$ and Hdh$^{Q7/Q111}$ mice treated with thioperamide. D$_1$R-H$_3$R heteromers were visualized in all samples as green spots around blue colored DAPI stained nucleus. Scale bar: 20 μm. In (E) the right panel, the number of cells containing one or more green spots is expressed as the percentage of the total number of cells (blue nucleus). *r values* (number of green spots/cell containing spots) are shown above each bar. Data (% of positive cells or r) are the mean ± SEM of counts in 600–800 cells from 4 to 8 different fields from three different animals. Student's *t* test showed no significant differences in heteromer expression in thioperamide-treated Hdh$^{Q7/Q111}$ mice compared to the respective Hdh$^{Q7/Q7}$ mice.

The online version of this article includes the following figure supplement(s) for figure 4:

**Figure supplement 1.** Functional D$_1$R-H$_3$R heteromers are expressed in 5-month-old Hdh$^{Q7/Q7}$ and Hdh$^{Q7/Q111}$ mice.

**Figure supplement 2.** Schematic representation of pharmacological treatments and behavioral analysis performed after chronic treatment with saline or thioperamide.

**Figure supplement 3.** No significant differences in the open field habituation were found between treatments and genotypes.

**Figure supplement 4.** Training session in the T-SAT showed similar number of arm entries in all genotypes and treatments.

**Figure supplement 5.** Expression of D$_1$R-H$_3$R heteromers in 8-month-old Hdh$^{Q7/Q7}$ and Hdh$^{Q7/Q111}$ mice chronically treated with thioperamide.

observed that in saline-treated 6-mo-old Hdh$^{Q7/Q111}$ mice the heteromer expression was significantly diminished with respect to the age-matched Hdh$^{Q7/Q7}$ mice (*Figure 3—figure supplement 1A,B*). Notably, treatment with thioperamide significantly prevented the loss of D$_1$R-H$_3$R heteromers in all brain regions analyzed in Hdh$^{Q7/Q111}$ mice at both 6 (*Figure 4D,E*) and 8 mo of age (*Figure 4—figure supplement 5A,B*).

## Treatment with thioperamide ameliorates spinophilin-immunoreactive puncta alterations in the motor cortex and hippocampus of 6-month-old mutant Hdh$^{Q7/Q111}$ mice

Alterations in dendritic spine dynamics, density and morphology are critically involved in the synaptic deficits present in HD (*Brito et al., 2014*; *Ferrante et al., 1991*; *Guidetti et al., 2001*; *Lynch et al., 2007*; *Milnerwood et al., 2006*; *Puigdellívol et al., 2015*; *Simmons et al., 2009*; *Sotrel et al., 1993*; *Spires et al., 2004*). We recently described a significant decrease in dendritic spine density in the hippocampus (*Brito et al., 2014*) and the motor cortex of mutant Hdh$^{Q7/Q111}$ mice (*Puigdellívol et al., 2015*) without significant alterations in the striatum. To analyze whether the improvement of motor learning and memory deficits observed in thioperamide-treated mutant Hdh$^{Q7/Q111}$ mice was associated with a recovery in the density of dendritic spines, spinophilin immunostaining was performed in CA1 hippocampal and motor cortical coronal slices obtained from 6-mo-old wild-type Hdh$^{Q7/Q7}$ and mutant Hdh$^{Q7/Q111}$ mice (*Figure 5A,B* and *Figure 5—figure supplement 1A*). This methodology was used by us and others to identify structural alterations in dendritic spines (*Hao et al., 2003*; *Puigdellívol et al., 2015*; *Tang et al., 2004*). Confocal microscopy analyses revealed a significant reduction in the density of spinophilin-immunoreactive puncta in the *stratum radiatum* (apical dendrites of CA1 pyramidal neurons) and *stratum oriens* (basal dendrites of CA1 pyramidal neurons) of saline-treated 6-mo-old mutant Hdh$^{Q7/Q111}$ mice compared to saline-treated wild-type Hdh$^{Q7/Q7}$ mice (*Figure 5A* and *Figure 5—figure supplement 1A*). Interestingly, thioperamide treatment prevented the decline in the number of spinophilin-immunoreactive puncta in mutant Hdh$^{Q7/Q111}$ mice (*Figure 5A* and *Figure 5—figure supplement 1A*). Similar data was obtained when the layers of the motor cerebral cortex (M1) were analyzed. A significant reduction in the density of spinophilin-immunoreactive puncta in layer I and layer II-III, but not layer V, of the motor cortex of 6-mo-old saline-treated Hdh$^{Q7/Q111}$ mice was found compared to saline-treated Hdh$^{Q7/Q7}$ mice (*Figure 5B* and *Figure 5—figure supplement 1A*). Interestingly, thioperamide-treated Hdh$^{Q7/Q111}$ mice exhibited a complete recovery in the density of spinophilin-immunoreactive puncta (*Figure 5B* and *Figure 5—figure supplement 1A*). No significant differences were found between groups when the mean size of spinophilin puncta was analyzed (*Figure 5—figure*

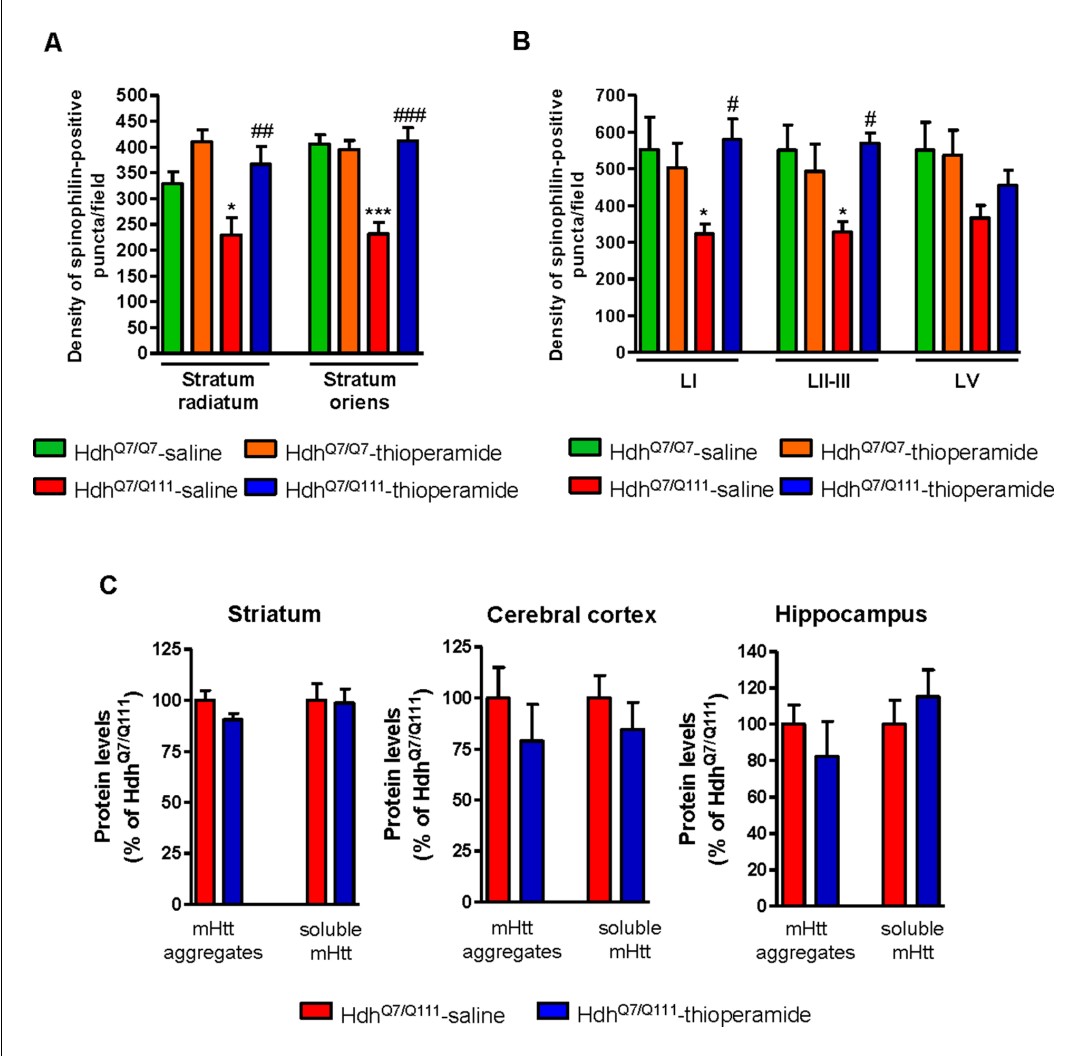

**Figure 5.** Thioperamide treatment restored spinophilin-immunoreactive puncta reduction in the hippocampus and motor cortex of Hdh[Q7/Q111] mice and exerts no effect on the clearance of mutant huntingtin accumulation. In (**A**) spinophilin-immunoreactive puncta were counted in the *stratum oriens* and *stratum radiatum* of CA1 hippocampus and in (**B**) layers I, II/III and V of motor cortex area 1 (M1) of saline and thioperamide-treated Hdh[Q7/Q7] and Hdh[Q7/Q111] mice. Quantitative analysis is shown as mean ± SEM (n = 9 images from three animals/group). Statistical analysis was performed using Student's two-tailed *t* test. *p<0.05, ***p<0.001 compared to saline-treated Hdh[Q7/Q7] mice. #p<0.05, ##p<0.01, ###p<0.001 compared to saline-treated Hdh[Q7/Q111] mice. In (**C**), Quantification of the protein levels of insoluble mHtt oligomeric forms and soluble mHtt forms of total striatal, hippocampal and cortical extracts from 6-month-old saline and thioperamide-treated Hdh[Q7/Q111] mice analysed by immunoblot. All histograms represent the mean ± SEM (n = 6–8 per group). Student's *t* test showed no significant differences between groups.

The online version of this article includes the following figure supplement(s) for figure 5:

**Figure supplement 1.** Biochemical and Pathological Effects of Thioperamide treatment.

*supplement 1A*). Altogether, these data demonstrate that the loss of spinophilin immunoreactive-puncta in mutant Hdh[Q7/Q111] mice can be ameliorated by thioperamide treatment.

We also evaluated mutant huntingtin (mhtt) aggregates in the striatum, cerebral cortex and hippocampus of mutant Hdh[Q7/Q111] mice after saline or thioperamide treatment, as another pathological hallmark of HD (*Arrasate and Finkbeiner, 2012*; *Giralt et al., 2012*; *Hoffner et al., 2007*). 1C2 immunostaining revealed in lysates from either vehicle or treated mutant Hdh[Q7/Q111] mice a substantial accumulation of mhtt oligomeric forms detected as a diffuse smear in the stacking gel (*Figure 5—figure supplement 1B*). Thioperamide treatment failed to prevent the accumulation of these

oligomeric forms (*Figure 5C* and *Figure 5—figure supplement 1B*). No significant differences between groups were found when soluble monomeric mhtt levels were analyzed (*Figure 5C* and *Figure 5—figure supplement 1B*).

## Thioperamide treatment does not rescue memory and motor learning deficits in mutant Hdh$^{Q7/Q111}$ mice when D$_1$R-H$_3$R heteromers are lost

If the behavioral improvements observed after thioperamide treatment are mediated by the D$_1$R-H$_3$R heteromer and not just by the blockade of the single H$_3$R, then a treatment paradigm in the absence of the heteromer should have no effect. To test this hypothesis, we used wild-type Hdh$^{Q7/Q7}$ and mutant Hdh$^{Q7/Q111}$ mice at the age of 7 months, when we found the heteromer to be lost. Animals were chronically treated with saline or thioperamide for 1 month and motor learning was evaluated using the accelerating rotarod task. As expected, saline-Hdh$^{Q7/Q111}$ mice exhibited poor performance in this task showing shorter latency to fall compared to wild-type Hdh$^{Q7/Q7}$ mice (*Figure 6A*). Notably, thioperamide treatment had no effect on motor learning performance as both saline- and thioperamide-treated mutant Hdh$^{Q7/Q111}$ mice were indistinguishable demonstrated by similar latency to fall in the accelerating rotarod task (*Figure 6A*).

We next asked whether thioperamide treatment could improve cognitive function by rescuing memory deficits in these same animals. Saline-treated 8-mo-old Hdh$^{Q7/Q111}$ mice exhibited long-term memory deficits when recognition memory was analyzed using the novel object recognition test (NORT) (*Figure 6B*). Similar to motor learning results, chronic treatment with thioperamide did not rescue Hdh$^{Q7/Q111}$ mice from memory deficits (*Figure 6B*). Overall, these results demonstrate that the effect of thioperamide in learning and memory in Hdh$^{Q7/Q111}$ mice requires the proper expression and function of D$_1$R-H$_3$R heteromers.

## D$_1$R-H$_3$R heteromer expression changes occur in other rodent HD models and in HD patients

The fact that thioperamide treatment 1) prevents cognitive and motor learning deficits, 2) ameliorates striatal neuropathology, 3) ameliorates morphological alterations and 4) prevents the loss of D$_1$R-H$_3$R heteromers at 6 mo and 8 mo of age in a mouse model of HD is suggestive that thioperamide, or a future pharmacologically improved H$_3$R antagonist specifically targeting D$_1$R-H$_3$R heteromers, can be used to treat HD symptoms. To test this, we investigated D$_1$R-H$_3$R heteromer expression in other transgenic HD mouse models and in human caudate-putamen slices using PLA. The loss of heteromer expression compared with wild-type littermates was also observed in other mouse models of HD, the R6/1 and R6/2 mice transgenic for the human huntingtin exon 1 (*Figure 7—figure supplement 1A,B*, respectively). Importantly, D$_1$R-H$_3$R heteromers were detected as green spots surrounding the blue stained nuclei in human caudate-putamen slices from control individuals and low-grade (grade 0, 1 and 2) HD patients (*Figure 7A,B*). In contrast, green spots were almost absent in samples from high-grade (grade 3 or grade 4) HD patients (*Figure 7A,B*). These results show that D$_1$R-H$_3$R heteromer formation changes during disease progression and, importantly, that humans express D$_1$R-H$_3$R heteromers at early disease stages.

## Discussion

The imbalance of dopamine inputs throughout HD progression represents a potential 'point of no return' for HD patients as this disequilibrium can eventually lead to substantial neuronal dysfunction and cell death. In the present study, we demonstrate that 1) excess dopamine signaling via D$_1$R leads to cell death by activating the p38 pathway; 2) D$_1$R-H$_3$R complexes are found within the striatum, cortex and hippocampus of WT mice and in HD mice at early but not late disease stages; 3) targeting D$_1$R via D$_1$R-H$_3$R complexes can slow progression of the disease in early but not late stages when the complexes are lost; and 4) D$_1$R-H$_3$R complexes are expressed in the human brain and thus represent potential therapeutic targets. This is the first demonstration of GPCR heteromers as potential targets to treat HD. Together, these data support a novel role for D$_1$R-H$_3$R complexes in neuroprotection and HD.

Several studies have revealed that dopamine neurotoxicity increases the sensitivity of MSSNs to glutamate inputs and leads to striatal neurodegeneration, a role ascribed to aberrant D$_1$R and not D$_2$R (*Cepeda and Levine, 1998*; *Flores-Hernández et al., 2002*; *Paoletti et al., 2008*; *Tang et al.,*

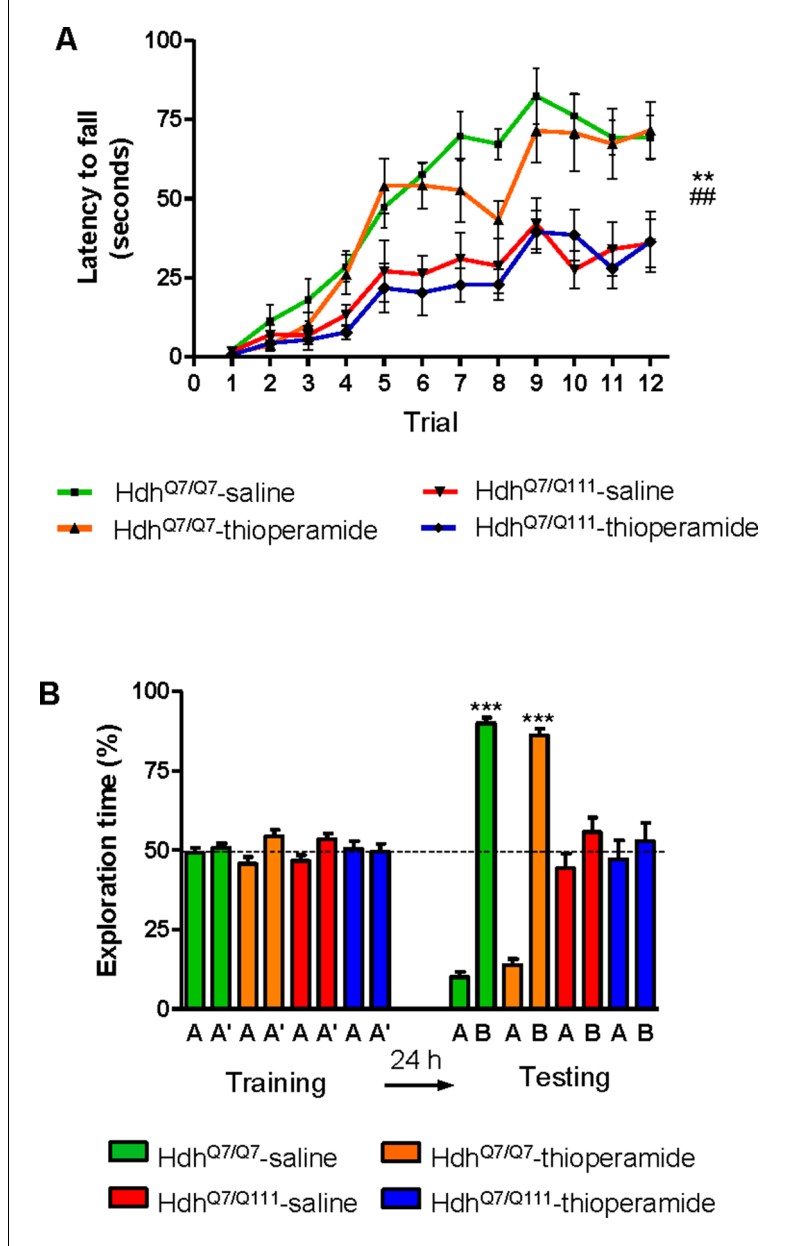

**Figure 6.** Thioperamide chronic treatment does not prevent motor learning and long-term memory (LTM) deficits in 8-month-old Hdh$^{Q7/Q111}$ mice when the D$_1$R-H$_3$R heteromer is not expressed. In (**A**), curves illustrating the latency to fall in the accelerating rotarod of 8-month-old Hdh$^{Q7/Q7}$ and Hdh$^{Q7/Q111}$ mice treated with saline or thioperamide from 7 months of age are shown. Two-way ANOVA with repeated measures showed significant differences (\*\*$p<0.01$) of saline-treated Hdh$^{Q7/Q111}$ mice compared to saline-treated Hdh$^{Q7/Q7}$ mice or (##$p<0.01$) thioperamide-treated Hdh$^{Q7/Q111}$ mice compared to saline-treated Hdh$^{Q7/Q7}$ mice. 11 saline-treated Hdh$^{Q7/Q7}$ mice, 11 thioperamide-treated Hdh$^{Q7/Q7}$ mice, eight saline-treated Hdh$^{Q7/Q111}$ mice and nine thioperamide-treated Hdh$^{Q7/Q111}$ mice were evaluated at 8 months of age. In (**B**), bar diagram illustrating the exploration time for saline or thioperamide-treated Hdh$^{Q7/Q7}$ and Hdh$^{Q7/Q111}$ mice during the training and the testing (24 hr delay, LTM) sessions in a novel-object recognition task showing that long-term recognition memory deficits are not rescued in the thioperamide-treated Hdh$^{Q7/Q111}$ mice. One-way ANOVA with Bonferroni *post hoc* comparisons showed significant differences (\*\*\*$p<0.001$) compared to the old object recognition. 11 saline-treated Hdh$^{Q7/Q7}$ mice, 12 thioperamide-treated Hdh$^{Q7/Q7}$ mice, 10 saline-treated Hdh$^{Q7/Q111}$ mice and 11 thioperamide-treated Hdh$^{Q7/Q111}$ mice were evaluated at 8 months of age.

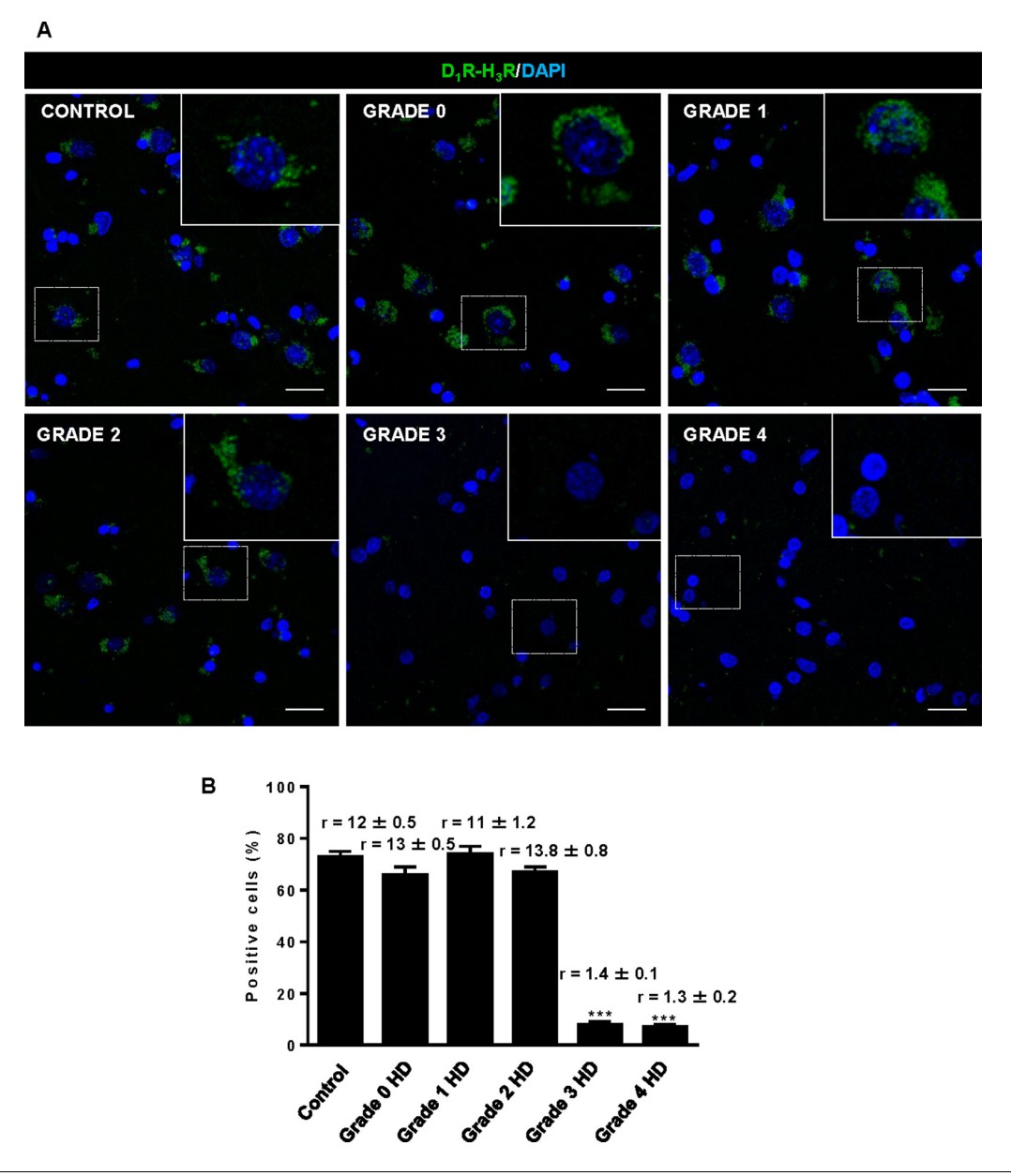

**Figure 7.** Striatal $D_1R$-$H_3R$ heteromers are expressed in human control subjects and grade 2 HD patients but not in grade 3–4 HD patients. In (**A**), by Proximity Ligation Assays (PLA), $D_1R$-$H_3R$ heteromers were visualized as green spots around blue colored DAPI stained nucleus in human striatal slices from age matched control subjects and 0–2 grade HD patients but not in 3–4 grade HD patients. Scale bar: 20 μm. In (**B**), the number of cells containing one or more green spots is expressed as the percentage of the total number of cells (blue nucleus). *r values* (number of green spots/cell containing spots) are shown above each bar. Data are mean ± SEM of counts in 600–800 cells from 10 different fields from subject described in Materials and Methods. Student's *t* test showed a significant (***p<0.001) decrease of heteromers expression in 3–4 grade HD patients compared to control subjects.

The online version of this article includes the following figure supplement(s) for figure 7:

**Figure supplement 1.** $D_1R$-$H_3R$ heteromer are not expressed in HD R6/1 and R6/2 mouse models.

*2007*). Thus, pharmacological treatments aimed to reduce $D_1R$ signaling may be beneficial to prevent or slow striatal cell death. Although we cannot rule out the participation of $D_2R$ in striatal degeneration, our results suggest that $D_1R$ is a major executor of the final signaling cascades that lead to cell death in HD. This is further supported by the fact that $D_1R$ is in excess over $D_2R$ in the striatum, so it is plausible that the former will be more significantly activated than the latter at increased DA levels. We have demonstrated that a toxic but not sub-toxic concentration of SKF81297 activates the p38 pro-apoptotic pathway, despite both concentrations triggering calcium release, albeit at different levels. Accordingly, p38 inhibitors completely abrogated the cell death induced by SKF81297 treatment, supporting the benefits of modulation of $D_1R$ signaling as potential treatment in HD. However, direct manipulation of DA production and/or $D_1R$ signalling via a specific antagonist has limited therapeutic ability due to associated deleterious side effects. An alternative approach is to modify $D_1R$ signalling via the histamine neuromodulator. An interaction between $H_3R$ and the dopaminergic system has been previously reported by us and others (*Kononoff Vanhanen et al., 2016*; *Rapanelli et al., 2016*; *Rapanelli et al., 2014*). In this frame, we have demonstrated that $H_3R$ ligands completely abrogate striatal cell death induced by $D_1R$, likely by inhibition of $D_1R$-mediated calcium influx and p38 activation. Importantly, $D_1R$-$H_3R$ complexes were found in the striatum, cortex and hippocampus from wild-type Hdh$^{Q7/Q7}$ and mutant Hdh$^{Q7/Q111}$ mice, regions known to be affected by mutant huntingtin toxicity (*Reiner et al., 1988*; *Rosas et al., 2003*; *Vonsattel and DiFiglia, 1998*).

The mechanisms of action of $D_1R$-$H_3R$ heteromers can be multiple including allosteric effects. Indeed, the efficacy of the disrupting peptides supports protein-protein-driven effects. A second and potentially additional mechanism is that heteromer formation may alter the trafficking of $D_1R$, which could have pleiotropic consequences on signaling. For example it is known that overstimulation of $D_1R$ induces receptor internalization promoting rapid intracellular signaling (*Kotowski et al., 2011*) and that receptor internalization can activate secondary signaling pathways (*Lohse and Calebiro, 2013*). We observe in vitro that thioperamide treatment maintains the PLA signal while in vivo we see similar effects. The signaling effects we observe appears to be on a variety of concentrations and timescales in agreement with previous studies showing that GPCR signaling occurs with varied kinetics (*Calebiro et al., 2010a*; *Calebiro et al., 2010b*). Indeed, part of the concern of trying to target GPCR heteromers for therapeutic purposes is the uncertainty around their stability and thus indirectly whether they can impact GPCR signaling at every timescale. For the case of $D_1R$-$H_3R$ heteromers, it appears that they are stable enough that they can affect both rapid receptor signaling (e.g., $Ca^{2+}$ mobilization) and longer cell signaling pathways like p38, two events that have previously been involved in neuronal cell death in HD (*Dau et al., 2014*; *Fan et al., 2012*; *Muller and Leavitt, 2014*; *Taylor et al., 2013*; *Wang et al., 2013*). It is unclear what controls $D_1R$-$H_3R$ heteromer formation or why it is lost during progression of HD. Whether it is a change in expression of an accessory protein, a post-translational modification or a change in cell physiology/morphology remains to be explored.

Our findings do not rule out that $H_3R$ ligands by targeting $D_2R$-$H_3R$ heteromers (*Ferrada et al., 2008*) could block $D_2R$ signaling and contribute to cell death protection. However, several findings argue in favor of $D_1R$-$H_3R$ heteromer as uniquely responsible for the effects of thioperamide on cell death reduction. First, $D_1R$ over-activation induces cell death-related pathways and $D_1R$-$H_3R$ disruption. In addition, pre-treatment with $H_3R$ ligands can block $D_1R$-induced cell death and prevent $D_1R$-$H_3R$ loss. Finally, the effect of TAT-peptide analogues of $D_1R$ transmembrane domains in $D_1R$-$H_3R$ stability and function demonstrate that we are observing specific $D_1R$-$H_3R$, and not $D_2R$-$H_3R$, signaling and function. Thioperamide has recently been suggested to act via the $H_4R$ receptor. However, several pieces of our data suggest $H_4R$ is not responsible for the observed effects. First, we measured similar effects using VUF 5681, a different $H_3R$ antagonist. In addition we lose all effects of thioperamide in cells where $H_3R$ expression was silenced or when $D_1R$-$H_3R$ heteromers are lost in the mice yet $H_4R$ should still be expressed. Finally, $H_4R$ is thought to be mainly expressed peripherally, while our data from brain slices and from mice which are predominatly cognitive in nature, strongly implicate the CNS.

Besides striatal and cortical cell death, growing evidence points to neuronal dysfunction as responsible for the earliest HD disturbances in cognitive and behavioral changes (*Lemiere et al., 2004*; *Puigdellívol et al., 2016*). Despite these early changes, no effective treatments are currently available to treat cognitive decline in HD. Moreover, the timing of intervention is also critical since

atrophy and dysfunction progress with age and treatment may be different according to the stage of illness. In this scenario, and given the well-known role of both dopamine and histamine in synaptic plasticity and memory (*Cahill et al., 2014*; *Ellender et al., 2011*; *Haas et al., 2008*; *Komater et al., 2005*; *López de Maturana and Sánchez-Pernaute, 2010*; *Mohsen et al., 2014*; *Orsetti et al., 2002*; *Pascoli et al., 2009*; *Wiescholleck and Manahan-Vaughan, 2014*), it is possible that the therapeutic potential of H₃R ligands as modulators of D₁R-H₃R heteromers could also be extended to improve learning impairments and cognitive decline in HD. This is supported by our data showing that chronic treatment with the H₃R antagonist thioperamide at 5 months of age prevented motor learning deficits, as well as impaired spatial and recognition memories in mutant Hdh$^{Q7/Q111}$ mice. Importantly, thioperamide treatment does not induce off-target effects (such as alterations in spontaneous locomotor activity or anxiety-like behaviors) neither in wild-type Hdh$^{Q7/Q7}$ nor in mutant Hdh$^{Q7/Q111}$ mice. In addition, early chronic treatment with thioperamide prevented disruption of the heteromer at 6 and 8 months of age and the subsequent cognitive decline. It seems unlikely that there is a direct link between D₁R-H₃R heteromers and cognitive deficits, but the data do suggest that whatever neuronal changes occur during progression of the disease they are blocked or at minimum delayed. Importantly, we can say that D₁R-H₃R heteromers are required for this effect as thioperamide treatment at 7 months of age (when the heteromer is lost in HD mice) is not able to prevent cognitive and motor learning deficits. This latter result might explain the results of the effects that GSK189254, an H₃R antagonist, have in a Q175 mouse model of HD (*Whittaker et al., 2017*). The authors saw no change in motor performance and mild improvement in exploratory behavior as measured in the Open Field test and in cognitive function as measured by a T-maze. Our data suggest that D₁R-H₃R heteromer expression is crucial to the efficacy of H₃R antagonists as a therapeutic option in HD.

What disease-driven neuronal changes are prevented by H₃R antagonism through the D₁R-H₃R heteromer is not completely clear. However, we did find that chronic thioperamide treatment at early stages completely rescue the reduction in the density of spinophilin-immunoreactive puncta in HD mice in both hippocampal and cortical areas, suggesting that adequate dopaminergic signaling is required for normal forms of synaptic structural plasticity and cognitive processes. Substantial data support the importance of dopamine receptors for synaptic plasticity in the cortex and hippocampus (*Levy and Goldman-Rakic, 2000*; *Robbins, 2000*; *Sajikumar and Frey, 2004*). In this view, any dopamine imbalance with both suboptimal and supra-optimal dopamine activity has been reported to modify cognitive performance (*Mattay et al., 2003*; *Vijayraghavan et al., 2007*). As the early stages of HD may reflect a hyperdopaminergic stage (*Chen et al., 2013a*; *Mochel et al., 2011*), treatments reducing dopamine signaling may have therapeutic benefits. In fact, dopamine-depleting drugs such as tetrabenazine or dopamine-stabilizers as pridopidine showed neuroprotective effects in HD mice (*Wang et al., 2010*), and improve motor coordination abnormalities in HD patients (*Huntington Study Group, 2006*; *de Yebenes et al., 2011*), while specific D₁R inhibition rescues electrophysiological changes in excitatory and inhibitory synaptic transmission in full-length HD mouse models (*André et al., 2011b*). However, none of these treatments have demonstrated cognitive improvements. The suggestion that D₁R-H₃R heteromers may be legitimate targets for the treatment of HD shines a spotlight on what continues to be an elusive drug target. Indeed, in the context of this study, the loss of the heteromer in disease progression despite the fact that the receptors themselves are still expressed and functional, points to the heteromers as optimal targets rather than the single receptors. The concept of heteromers have been known for over a decade but physiologic examples have only recently come to be appreciated (*Bonaventura et al., 2015*; *Viñals et al., 2015*; *Baba et al., 2013*; *Fribourg et al., 2011*; *González et al., 2012a González et al., 2012b*; *Kern et al., 2012*; *Navarro et al., 2015*). In sum, our study showing that H₃R antagonists can prevent learning and memory deficits by blocking D₁R in D₁R-H₃R complexes, along with the role of these heteromers on neuronal cell death, predict a critical role of the histaminergic system as modulator of the dopamine imbalance in HD, and may help to overcome the deleterious effects of directly manipulating DA-production and/or signaling, thus opening new and important alternatives for HD therapeutics.

# Materials and methods

**Key resources table**

| Reagent type (species) or resource | Designation | Source or reference | Identifiers | Additional information |
|---|---|---|---|---|
| Cell line (*H. sapiens*) | HEK293 (Human embryonic kidney293 cells) | American Type Culture Collection | | |
| Cell line (*M. musculus*) | STHdh$^{Q7}$; STHdh$^{Q111}$ (mouse striatal neuronal progenitor cells) | Dr M Macdonald (Center for Genomic Medicine, Boston, USA) | | |
| Strain, strain background (*Mus musculus*) | Hdh$^{Q7/Q111}$; Hdh$^{Q7/Q7}$ | Dr M Macdonald (Center for Genomic Medicine, Boston, USA) | Hdh$^{Q111}$ MGI:1861935 | |
| Strain, strain background (*Mus musculus*) | R6/1; R6/2 | The Jackson Laboratory (Bar Harbor, ME, USA) | R6/1: MGI:2389466 For R6/2: MGI:2386951 | |
| Strain, strain background (*H. sapiens*) | *Post-mortem* human brain sections containing caudate-putamen | Tissue Bank at Hospital Universitario Fundación Alcorcón (Madrid, Spain) Netherlands Brain Bank (Amsterdam, The Netherlands) | | For details and characteristics of human samples see: "Moreno E., et al., Neuropsychopharmacology. 2018 PMID:28102227' |
| Antibody | anti-D$_1$R (guinea pig) | Frontier Institute | Cat. # D-1R-GP-Af500 RRID:AB_2571595 | Dilution: 1/200; 1/100 |
| Antibody | anti-H$_3$R (rabbit polyclonal) | Alpha diagnostic | Cat. # H3R31-A RRID:AB_1617140 | Dilution: 1/200 |
| Antibody | goat Alexa Fluor 488 anti-guinea pig antibody | Jackson Immunoresearch Laboratories | Cat. #106-545-003 RRID:AB_2337438 | Dilution: 1/100 |
| Antibody | anti-phospho-p38 MAPK (Thr180/Tyr182) (rabbit polyclonal) | Cell Signaling | Cat. #9211S RRID:AB_331641 | Dilution: 1/1,000 |
| Antibody | anti-β-tubulin (mouse monoclonal) | Sigma | Cat# SAB4200715 RRID:AB_2827403 | Dilution: 1/10,000 |
| Antibody | IRDye 680 goat anti-rabbit antibody | Li-cor | Cat. #926–68071 RRID:AB_10956166 | Dilution: 1/10,000 |
| Antibody | IRDye 800 goat anti-mouse antibody | Li-cor | Cat. # 926–32210 RRID:AB_621842 | Dilution: 1/10,000 |
| Antibody | anti-spinophilin (rabbit polyclonal) | Millipore | Cat# 06–852 RRID:AB_310266 | Dilution: 1/250 |
| Antibody | Cy3 anti-rabbit secondary antibodies | Jackson Immuno Research Laboratories | Cat# 111-165-003 RRID:AB_2338000 | Dilution: 1/200 |
| Antibody | Anti-1C2 (mouse monoclonal) | Millipore | Cat# MAB1574 RRID:AB_94263 | Dilution: 1/1,000 |
| Recombinant DNA reagent | Clone V3LHS_638095 | Thermo Scientific | | |
| Recombinant DNA reagent | Clone V3LHS_638091 | Thermo Scientific | | |
| Recombinant DNA reagent | psPAX2 | Addgene#12260 | | |
| Recombinant DNA reagent | pMD2.G | Addgene#12259 | | |
| Recombinant DNA reagent | RHS4346 | Thermo Scientific | | |

*Continued on next page*

*Continued*

| Reagent type (species) or resource | Designation | Source or reference | Identifiers | Additional information |
|---|---|---|---|---|
| Recombinant DNA reagent | H₃R-shRNA and control-shRNA | This study | | See Materials and methods |
| Sequence-based reagent | RT-qPCR primers | This study | | See Materials and methods |
| Peptide, recombinant protein | TAT-TM peptides | This study | | See Materials and methods |
| Commercial assay or kit | Duolink II in situ PLA detection reagent red Kit | Sigma | Cat. #DUO92008 | |
| Commercial assay or kit | Duolink II PLA probe anti-guinea pig minus | Sigma | Cat. #DUO92010 | |
| Commercial assay or kit | Duolink II PLA probe anti-rabbit plus | Sigma | Cat. #DUO92002 RRID:AB_2810940 | |
| Commercial assay or kit | High Capacity cDNA Reverse Transcription Kit | Applied Biosystems | Cat. #4368814 | |
| Commercial assay or kit | Amplified Luminiscent Proximity Homogeneous Assay kit | AlphaScreen *SureFire* p-ERK 1/2 (Thr202/Tyr204) Assay Kits PerkinElmer | Cat. # TGRESB | |
| Commercial assay or kit | [³H] SCH 23390 | PerkinElmer | Cat. # NET930 | 0.02 nM to 10 nM |
| Commercial assay or kit | [³H] R-α-methyl histamine | Perkinelmer | Cat. # NET1027 | 0.1 nM to 20 nM |
| Commercial assay or kit | SB 203580 | Tocris | Cat. # 1402 | 1 µM; 10 µM (see Materials and methods) |
| Commercial assay or kit | SKF 81297 | Tocris | Cat. # 1447 | 100 nM; 1 µM; 30 µM; 50 µM (see Materials and methods) |
| Commercial assay or kit | SCH 23390 | Tocris | Cat. # 0925 | one to 50 µM (see Materials and methods) |
| Commercial assay or kit | Thioperamide maleate salt | Sigma-Aldrich | Cat. #T123 | 10 µM (cells) 10 mg/kg (mice) |
| Software, algorithm | Grafit | Erithacus (http://www.erithacus.com/grafit/) | | |
| Software, algorithm | ImageJ | ImageJ (https://imagej.nih.gov/ij/) | RRID:SCR_003070 | |
| Software, algorithm | SMART junior | Panlab (http://www.panlab.com/panlabWeb/Software/php/displaySoft.php?nameSoft=SMART JUNIOR) | RRID:SCR_012154 | |
| Software, algorithm | GraphPad Prism | GraphPad Prism (https://www.graphpad.com/) | RRID:SCR_015807 | Version 6 |

## Human brain slices

Paraffin-embedded *post-mortem* 4 µm-thick brain sections containing caudate-putamen were obtained and provided by the Tissue Bank at Hospital Universitario Fundación Alcorcón (Madrid, Spain) and the Netherlands Brain Bank (Amsterdam, The Netherlands) according to the standardized procedures of both institutions. The samples analyzed were from patients with HD (1 grade 0; 1 grade 1; 2 grade 2; 3 grade 3 and 3 grade four patients) and from age matched controls with no neurological disease (three subjects). All protocols were approved by the institutional ethic committees.

## Cell cultures

Mouse striatal wild-type STHdh$^{Q7}$ and mutant STHdh$^{Q111}$ cell lines were provided by Dr M. Macdonald (Center for Genomic Medicine, Boston, USA) and confirmed by PCR. These conditionally immortalized wild-type STHdh$^{Q7}$ and mutant STHdh$^{Q111}$ striatal neuronal progenitor cell lines expressing endogenous levels of normal and mutant huntingtin with 7 and 111 glutamines, respectively, have been described previously (*Trettel et al., 2000*). These cells do not exhibit amino-terminal inclusions allowing the study of changes involved in early HD pathogenesis (*Trettel et al., 2000*). Striatal cells were grown at 33˚C in DMEM (Sigma-Aldrich), supplemented with 10% fetal bovine serum (FBS), 1% streptomycinpenicillin, 2 mM L-glutamine, 1 mM sodium pyruvate, and 400 g/ml G418 (Geneticin; Invitrogen).

HEK293 cells were purchased from ATCC and kept below passage 20. Cells were grown in Dulbecco's modified Eagle's medium (DMEM) (Gibco, Paisley, Scotland, UK) supplemented with 2 mM L-glutamine, 100 µg/ml sodium pyruvate, 100 U/ml penicillin/streptomycin, essential medium nonessential amino acids solution (1/100) and 5% (v/v) heat inactivated fetal bovine serum (Invitrogen, Paisley, Scotland, UK) and were maintained at 37˚C in an atmosphere with 5% $CO_2$. Cells were transiently transfected with the corresponding fusion protein cDNA using Lipofectamine 3000 (Invitrogen, Paisley, Scotland, UK). Both cell lines were routinely test for mycoplasma contamination monthly by PCR.

## Animal models of HD

Knock-in mice, with targeted insertion of 109 CAG repeats that extends the glutamine segment in murine huntingtin to 111 residues, and the corresponding littermates having seven glutamine residues were maintained on a C57BL/6 genetic background (*Lloret et al., 2006*). Hdh$^{Q7/Q111}$ heterozygous males and females were intercrossed to generate age-matched Hdh$^{Q7/Q111}$ heterozygous and Hdh$^{Q7/Q7}$ wild-type littermates. Only males were used for all experiments. Hemizigous male mice transgenic for exon 1 of the human huntingtin gene with a greatly expanded CAG repeat (~115 CAG repeats in R6/1 mice and ~160 CAG repeats in R6/2 mice) (*Mangiarini et al., 1996*) and wild-type littermates were used when indicated in proximity ligation assays. Animals were housed under a 12 hr light/dark cycle with food and water ad libitum.

## Mouse brain slices preparation

For PLA experiments, 2-, 4-, 6- and 8-month-old Hdh$^{Q7/Q7}$ and Hdh$^{Q7/Q111}$ mice were deeply anesthetized and immediately perfused transcardially with saline (PBS) followed by 4% paraformaldehyde (PFA)/phosphate buffer. Brains were removed and post-fixed overnight in the same solution, cryoprotected by immersion in 10, 20, 30% gradient sucrose (24 hr for each sucrose gradient) at 4˚C and then frozen in dry ice-cooled methylbutane. Serial coronal cryostat sections (30 µm) through the whole brain were collected in PBS-0.025% azide as free-floating sections and stored at 4˚C until PLA experiments were performed. For cell death determination, Hdh$^{Q7/Q111}$ and Hdh$^{Q7/Q7}$ mice were killed by cervical dislocation at the age of 4, 5 and 8 months. Mouse brains were rapidly removed and placed in ice-cold oxygenated ($O_2$/$CO_2$: 95%/5%) Krebs-HCO$_{3}^{-}$ buffer (124 mM NaCl, 4 mM KCl, 1.25 mM NaH$_2$PO$_4$, 1.5 mM MgSO$_4$, 1.5 mM CaCl$_2$, 10 mM glucose and 26 mM NaHCO$_3$, pH 7.4). Cerebral hemisferes were split and sliced coronally using a McIlwain chopper (Ted Pella, Inc, California) in sterile conditions. Striatum, cortex and hippocampal slices (300 µm thick) were kept at 4˚C in Krebs-HCO$_{3}^{-}$ buffer during the dissection and transferred into a Millicell Insert (Millipore).

## Cell death determination in striatal cells and in mouse organotypic slice cultures

Striatal STHdh$^{Q7}$ or STHdh$^{Q111}$ cells were grown to reach 50% of confluence on 12-well plates containing 3 cm$^2$-glass coverslips. Medium was then replaced by a new supplemented medium containing 0.5% FBS. Vehicle, SCH 23390, thioperamide or SB 203580 were added at the indicated concentrations to cells and incubated for 1 hr before the addition of D$_1$R. When TAT-TM peptides were applied to cell cultures, these were added 4 hr before the addition of D$_1$R agonist. After agonist addition, an additional incubation period of 24 hr was performed. Then cells were washed twice in cold-PBS and fixed with 4% paraformaldehyde for 1 hr at 4˚C. Sample nuclei were stained with Hoechst 1:1000. Stained cells were then washed with PBS and mounted under glass coverslips with

Mowiol. A minimum of 10 fields were taken from each coverslip using a fluorescence microscope and the plugin Image-based Tool for Counting Nuclei for ImageJ was used for the quantification of the total nuclei. In mouse organotypic cultures, brain slices (300 µm thickness, see above) were cultured for 24 hr into a Millicell Insert in Neurobasal medium supplemented with 20% horse serum, 0.5% B27, 2 mM L-glutamine, 100 µg/ml sodium pyruvate, non-essential amino acids solution (1/100) and 100 units/ml penicillin/streptomycin (all supplements were from Invitrogen, Paisley, Scotland, UK) before replacing with fresh medium. Vehicle, SCH 23390, thioperamide were added at the indicated concentrations to organotypic cultures and incubated for 1 hr before the addition of $D_1R$ agonist. TAT-TM peptides were applied to cell cultures 4 hr before the addition of $D_1R$ agonist. After agonist addition, an additional incubation period of 48 hr was performed. Then, 10 µM propidium iodide (PI) was added to organotypic cultures and maintained at 37°C for 1 hr. Organotypic cultures were washed twice in cold-PBS and fixed with 4% paraformaldehyde for 1 hr at 4°C. Total nuclei were stained with Hoechst 1:1000. The Hoechst stained and PI positive nuclei in organotypic cultures were counted to evaluate cell death in the brain slices. Quantification was performed using Leica SP2 confocal microscope (20x; UV, 561 lasers) and the quantification performed with the program Image-based Tool for Counting Nuclei for ImageJ. Cell death is expressed as the percentage of PI positive cells in the total Hoechst-stained nuclei.

## Lentivirus production and cell transduction

Silencing lentiviral vectors were produced by co-transfecting HEK293 producing cellsT with lentiviral silencing plasmids GIPZ Human histamine H3 receptor shRNA (Clone V3LHS_638095 or Clone V3LHS_638091, Thermo Scientific) with packing plasmid psPAX2 and envelope coding plasmid pMD2.G (Addgene#12260 and #12259, respectively) using the calcium phosphate method. For production of control non silencing lentiviral particles the $H_3R$ silencing plasmid were substituted with GIPZ Non-silencing Lentiviral shRNA Control (RHS4346, Thermoscientific). Infectious lentiviral particles were harvested at 48 hr post-transfection, centrifuged 10 min at 900 g to get rid of cell debris, and then filtered through 0.45 µm cellulose acetate filters. The titer of recombinant lentivirus was determined by serial dilution on HEK293T cells. For lentivirus transduction, striatal cells were subcultured to 50% confluence, cells were transduced with $H_3R$-shRNA-expressing lentivirus obtained with plasmid (Clone V3LHS_638095) or control-shRNA-expressing lentivirus (LV control) at a multiplicity of infection (MOI) of 10 in the presence of polybrene 5 µg/ml. Virus-containing supernatant was removed after 3 hr. Puromycin was added to the culturing media at the final concentration of 1 µg/ml 2 days after infection. 5 days after puromycin selection cells were transduced with the second $H_3R$-shRNA-expressing lentivirus obtained with plasmid Clone V3LHS_638091 to improve the level of silencing achieved. LV control infected cells were re-infected with control-shRNA-expressing lentivirus. The second infection was carried out as the first one. Cells were tested 72 hr after the second transduction was performed.

## RNA and real-time PCR

RNA was extracted using TRIzol Reagent (Molecular Research Center). 10 µg of total RNA were treated with RQ1 RNAse free DNAse (Promega) according to manufacturer instruction. DNAse treated DNA was quantified again and cDNA was synthesized using 2 µg total RNA with a High Capacity cDNA Reverse Transcription Kit; (Applied Biosystems). The mRNAs of actin, H3R and D1R were amplified by real-time (RT)-PCR using 1 µL cDNA and power SYBER green PCR Master Mix (Applied Biosystems) on a 7500 Real Time PCR system (Applied Biosystems). Primer sequences are as follows: MsACT For: ATGAGCTGCCTGACGGCCAGGTCAT, MsACT Rev: TGGTACCACCAGA-CAGCAC TGTGTT, $H_3R$ For: GCAACGCGCTGGTCATGCTC, $H_3R$ Rev: CCCCGGCCAAAGG TCCAACG, $D_1R$ FOR: ACCTCTGTGTGATCAGCGTG, AND $D_1R$ REV: GCGTATGTCCTGCTCAACC T. Thermal cycling conditions for amplification were set at 50°C for 2 min and 95°C for 10 min, respectively. PCR denaturing was set at 95°C for 15 s and annealing/extending at 60°C for 60 s for 40 cycles. mRNA levels normalized for actin are expressed as fold change relative to control cells. The results were quantified with the comparative $C_t$ method (known as the $2^{-\delta\delta Ct}$ method).

## In Situ Proximity Ligation Assays (PLA)

Cells or mouse or human brain slices were mounted on glass slides and treated or not with the indicated concentrations of receptor ligands or TAT-TM peptides for the indicated time. Then, cells or slices were thawed at 4°C, washed in 50 mM Tris-HCl, 0.9% NaCl pH 7.8 buffer (TBS), permeabilized with TBS containing 0.01% Triton X-100 for 10 min and successively washed with TBS. Heteromers were detected using the Duolink II in situ PLA detection Kit (OLink; Bioscience, Uppsala, Sweden) following the instructions of the supplier. A mixture of equal amounts of the primary antibodies: guinea pig anti-$D_1R$ antibody (1/200 Frontier Institute, Ishikari, Hokkaido, Japan) and rabbit anti-$H_3R$ antibody (1:200, Alpha diagnostic, San Antonio, Texas, USA) were used to detect $D_1R$-$H_3R$ heteromers together with PLA probes detecting guinea pig or rabbit antibodies, Duolink II PLA probe anti-guinea pig minus and Duolink II PLA probe anti-rabbit plus. Then samples were processed for ligation and amplification with a Detection Reagent Red and were mounted using a DAPI-containing mounting medium. Samples were observed in a Leica SP2 confocal microscope (Leica Microsystems, Mannheim, Germany) equipped with an apochromatic 63X oil-immersion objective (N.A. 1.4), and a 405 nm and a 561 nm laser lines. For each field of view a stack of two channels (one per staining) and 9 to 15 Z stacks with a step size of 1 μm were acquired. For PLA with brain slices, after image processing, the red channel was depicted in green color to facilitate detection on the blue stained nucleus and maintaining the color intensity constant for all images. A quantification of cells containing one or more spots versus total cells (blue nucleus) and, in cells containing spots, the ratio r (number of red spots/cell containing spots) were determined, using the Fiji package (http://pacific. mpi-cbg.de/), considering a total of 600–800 cells from 4 to 10 different fields within each brain region from three different mice per group or from three human control subjects, 3 human grade 3 or grade 4 HD patients, 2 grade 0 or grade 1 HD patients or 1 grade 2 HD patient. Nuclei and spots were counted on the maximum projections of each image stack. After getting the projection, each channel was processed individually. The nuclei were segmented by filtering with a median filter, subtracting the background, enhancing the contrast with the Contrast Limited Adaptive Histogram Equalization (CLAHE) plug-in and finally applying a threshold to obtain the binary image and the regions of interest (ROI) around each nucleus. Red spots images were also filtered and thresholded to obtain the binary images. Red spots were counted in each of the ROIs obtained in the nuclei images.

## Membrane preparation and radioligand binding

Striatal cells or mouse striatal, cortical or hippocampal tissue were homogenized in 50 mM Tris-HCl buffer, pH 7.4, containing a protease inhibitor mixture (1/1000, Sigma). The cellular debris was removed by centrifugation at 13,000 g for 5 min at 4°C, and membranes were obtained by centrifugation at 105,000 g for 1 hr at 4°C. Membranes were washed three more times at the same conditions before use. Ligand binding was performed with membrane suspension (0.2 mg of protein/ml) in 50 mM Tris–HCl buffer, pH 7.4 containing 10 mM $MgCl_2$, at 25°C. To obtain saturation curves, membranes were incubated with increasing free concentrations of [$^3$H] SCH 23390 (0.02 nM to 10 nM, PerkinElmer, Boston, MO, USA) or [$^3$H] R-$\alpha$-methyl histamine (0.1 nM to 20 nM, PerkinElmer, Boston, MO, USA) providing enough time to achieve stable equilibrium for the lower ligand concentrations. Nonspecific binding was determined in the presence of 30 μM non-labeled ligand. Free and membrane bound ligand were separated by rapid filtration of 500 μl aliquots in a cell harvester (Brandel, Gaithersburg, MD, USA) through Whatman GF/C filters embedded in 0.3% polyethylenimine that were subsequently washed for 5 s with 5 ml of ice-cold Tris–HCl buffer. The filters were incubated overnight with 10 ml of Ecoscint H scintillation cocktail (National Diagnostics, Atlanta, GA, USA) at room temperature and radioactivity counts were determined using a Tri-Carb 1600 scintillation counter (PerkinElmer, Boston, MO, USA) with an efficiency of 62%. Protein was quantified by the bicinchoninic acid method (Pierce Chemical Co., Rockford, IL, USA) using bovine serum albumin dilutions as standard. Monophasic saturation curves were analyzed by non-linear regression, using the commercial Grafit software (Erithacus Software), by fitting the binding data to the equation previously deduced (equation (3) in *Gracia et al., 2013*.

## Immunocytochemistry

Cells (60% confluence) were treated with vehicle or 30 µM SKF 81297 and after 45 min cells were kept at 4°C to block endocytosis/exocytosis, washed twice in cold-PBS, fixed in 4% paraformaldehyde for 15 min and washed with PBS containing 20 mM glycine (buffer A) to quench the aldehyde groups. After permeabilization with buffer A containing 0.05% Triton X-100 for 5 min, cells were washed with buffer A containing 1% bovine serum albumin (blocking solution) for 1 hr and labeled with the primary guinea pig anti-$D_1R$ antibody (1/100, Frontier Institute, Ishikari, Hokkaido, Japan, ON at 4°C), washed with blocking solution, and stained with the secondary goat Alexa Fluor 488 anti-guinea pig antibody (1:100, Jackson Immunoresearch Laboratories, West Grove, PA, USA, 2 hr at RT). Samples were washed twice with blocking solution, once with buffer A and finally with PBS. Nuclei were stained with 1:1000 Hoechst. Cells were mounted with Mowiol and observed in a Leica SP2 confocal microscope.

## Signaling in striatal cells

To determine ERK1/2 phosphorylation, cells (35,000/well) were cultured with a non-supplemented medium overnight before pre-treated at 25°C for 20 min with the antagonists and stimulated for an additional 7 min with the indicated agonists. Phosphorylation was determined by alpha-screen bead-based technology using the Amplified Luminescence Proximity Homogeneous Assay kit (PerkinElmer, Waltham, MA, USA) and the Enspire Multimode Plate Reader (PerkinElmer) following the instructions of the supplier. To determine calcium release, striatal cells were transfected with 4 µg of GCaMP6 calcium sensor (*Chen et al., 2013b*) using lipofectamine 3000. After 48 hr, cells were incubated (0.2 mg of protein/ml in 96-well black, clear bottom microtiter plates) with $Mg^{+2}$-free Locke's buffer pH 7.4 (154 mM NaCl, 5.6 mM KCl, 3.6 mM $NaHCO_3$, 2.3 mM $CaCl_2$, 5.6 mM glucose and 5 mM HEPES) supplemented with 10 µM glycine. When TAT-TM peptides treatment was performed they were added 1 hr before the addition of receptor ligands at the indicated concentration. Fluorescence emission intensity of GCaMP6s was recorded at 515 nm upon excitation at 488 nm on an EnSpire Multimode Plate Reader (PerkinElmer, Boston, MO, USA) for 330 s every 5 s and 100 flashes per well. The fluorescence gain was defined as a delta function of $\Delta F/F(t) = (F(t) – F0)/F0$, where F0 is the average fluorescence intensity in the first six measures from the start of recording and F(t) is the fluorescence intensity at a given time and was expressed in %. To determine p38 phosphorylation, striatal cells (80% confluence) were cultured with a non-supplemented medium 4 hr before the addition of the indicated ligand concentration for the indicated time and were lysed with 50 mM Tris-HCl pH 7.4, 50 mM NaF, 150 mM NaCl, 45 mM β-glycerophosphate, 1% Triton X-100, 20 µM phenyl-arsine oxide, 0.4 mM $NaVO_4$ and protease inhibitor cocktail. Lysates (20 µg protein) were processed for western blot a mixture of a rabbit anti-phospho-p38 MAPK (Thr180/Tyr182) antibody (1:1000, Cell Signaling) and a mouse anti-β-tubulin antibody (1:10,000, Sigma). Bands were visualized by the addition of a mixture of IRDye 680 anti-rabbit antibody (1:10,000, LI-COR Biosciences) and IRDye 800 anti-mouse antibody (1:10,000, LI-COR Biosciences) for 2 hr at room temperature and scanned by the Odyssey infrared scanner (LI-COR Biosciences). Band densities were quantified using the Odyssey scanner software. The level of phosphorylated p38 MAPK was normalized for differences in loading using the β-tubulin band intensities.

## Mice thioperamide treatment

Thioperamide maleate salt (Sigma-Aldrich, St. Louis, USA) was prepared fresh daily being dissolved in sterile 0,9% saline (NaCl) in order to deliver a final dose of 10 mg/kg in a final volume of 0.01 ml/g of body weight, as previously described (*Charlier et al., 2013*). The vehicle treatment consisted of an equal volume of saline solution. All injections were given via the intra-peritoneal route (*i.p*). Three *i.p* injections per week were administered to wild-type $Hdh^{Q7/Q7}$ and mutant knock-in $Hdh^{Q7/Q111}$ mice from 5 months of age until 6 months of age (when one cohort of animals was perfused to analyze PLA after behavioral assessment) or until 8 months of age (when a second cohort of animals were perfused to analyze PLA at this more advanced disease stage). A total of 11 saline-$Hdh^{Q7/Q7}$ mice, 10 thioperamide-$Hdh^{Q7/Q7}$ mice, seven saline-$Hdh^{Q7/Q111}$ mice and nine thioperamide-$Hdh^{Q7/Q111}$ mice were treated. For these experiments, a total of 11 saline-$Hdh^{Q7/Q7}$ mice, 10 thioperamide-$Hdh^{Q7/Q7}$ mice, seven saline-$Hdh^{Q7/Q111}$ mice and nine thioperamide-$Hdh^{Q7/Q111}$ mice were treated. Similarly, three *i.p* injections per week were administered to wild-type $Hdh^{Q7/Q7}$ and mutant knock-

in Hdh$^{Q7/Q111}$ mice from 7 months of age until 8 months of age to perform the behavioral studies when the D$_1$R-H$_3$R heteromers were lost. For these experiments, a total of 11 saline-Hdh$^{Q7/Q7}$ mice, 12 thioperamide-Hdh$^{Q7/Q7}$ mice, 10 saline-Hdh$^{Q7/Q111}$ mice and 11 thioperamide-Hdh$^{Q7/Q111}$ mice were treated. All treatments were performed in the afternoon to avoid the stress caused by the treatments during the behavioral assessment. Thus, during behavioral analysis treatments were performed after the evaluation of motor learning or cognitive tasks.

## Behavior assays

Accelerating rotarod was performed as previously described (*Puigdellívol et al., 2015*). Animals were placed on a motorized rod (30 mm diameter). The rotation speed gradually increased from 4 to 40 rpm over the course of 5 min. The time latency was recorded when the animal was unable to keep up on the rotarod with the increasing speed and fell. Rotarod training/testing was performed as four trials per day during three consecutive days. A resting period of one hour was left between trials. The rotarod apparatus was rigorously cleaned with ethanol between animal trials in order to avoid odors.

For T-maze spontaneous alternation task (T-SAT), the T-maze apparatus used was a wooden maze consisting of three arms, two of them situated at 180° from each other, and the third, representing the stem arm of the T, situated at 90° with respect to the other two. All arms were 45 cm long, 8 cm wide and enclosed by a 20 cm wall. Two identical guillotine doors were placed in the entry of the arms situated at 180°. In the training trial, one arm was closed (new arm) and mice were placed in the stem arm of the T (home arm) and allowed to explore this arm and the other available arm (old arm) for 10 min, after which they were returned to the home cage. After 5 hr (LTM), mice were placed in the stem arm of the T-maze and allowed to freely explore all three arms for 5 min. The arm preference was determined by calculating the time spent in each arm x 100/time spent in both arms (old and new arm). The T-maze was rigorously cleaned with ethanol between animal trials in order to avoid odors.

Novel object recognition test (NORT) consisted in a white circular arena with 40 cm diameter and 40 cm high. Mice were first habituated to the open field arena in the absence of objects (2 days, 15 min/day). During these two days of habitation, several parameters were measured to ensure the proper habituation of all mice in the new ambient. As a measure of anxiety or motivation behaviors, the distance that each mice rove in the periphery or in the center of the open field arena was measured as the rove distance in the periphery or in the center x 100/the total distance. The same analysis was performed by counting the number of entries in the periphery and in the center as well as the time that each mouse spent exploring the periphery or the center. The total distance that each mice rove during these two days of habituation was also recorded as a measure to evaluate spontaneous locomotor activity. On the third day, two similar objects were presented to each mouse during 10 min (A, A' condition) after which the mice were returned to their home cage. Twenty-four hours later (LTM), the same animals were re-tested for 5 min in the arena with a familiar and a new object (A, B condition). The object preference was measured as the time exploring each object ×100/time exploring both objects. The arena was rigorously cleaned with ethanol between animal trials in order to avoid odors. Animals were tracked and recorded with SMART junior software (Panlab, Spain).

## Immunohistochemistry, confocal microscopy and immunofluorescence-positive puncta counting

Saline and thioperamide-treated heterozygous mutant Hdh$^{Q7/Q111}$ and WT Hdh$^{Q7/Q7}$ mice at 6 months of age (n = 3 per group) were deeply anesthetized and immediately perfused transcardially with saline followed by 4% paraformaldehyde (PFA)/phosphate buffer. Brains were removed and postfixed overnight in the same solution, cryoprotected by immersion in 30% sucrose and then frozen in dry ice-cooled methylbutane. Serial coronal cryostat sections (30 μm) through the whole brain were collected in PBS as free-floating sections. Sections were rinsed three times in PBS and permeabilized and blocked in PBS containing 0.3% Triton X-100% and 3% normal goat serum (Pierce Biotechnology, Rockford, IL) for 15 min at room temperature. The sections were then washed in PBS and incubated overnight at 4°C with Spinophilin (1:250, Millipore) antibody that were detected with Cy3 anti-rabbit secondary antibodies (1:200, Jackson ImmunoResearch, West Grove, PA). As

negative controls, some sections were processed as described in the absence of primary antibody and no signal was detected. Confocal microscopy analysis and immunofluorescence-positive puncta counting spinophilin-positive spine-like structures was examined as previously described (*Puigdellívol et al., 2015*). Briefly, the images were acquired with Zeiss LSM510 META confocal microscope with HeNe lasers. Images were taken using a $\times$ 63 numerical aperture objective with $\times 4$ digital zoom and standard (one Airy disc) pinhole. Three coronal sections (30 μm thick) per animal (n = 3 per group) spaced 0.24 mm apart containing the motor area M1 or CA1 hippocampus were used. For each slice, we obtained three fields/cortical layer (I, II/III and V) of the M1 area and three fields/CA1 hippocampus (*stratum oriens* and *stratum radiatum*). The number and area of spinophilin-positive puncta were measured using NIH ImageJ version 1.33 by Wayne Rasband (National Institutes of Health, Bethesda, MD). To analyze spinophilin immunolabeling, brightness and contrast of fluorescence images were adjusted so that only punctate fluorescence, but no weak diffuse background labeling was visible. In the article, we use the term 'puncta' and 'cluster' interchangeable to refer to discrete points of protein at the fluorescence microscope. Positive puncta/cluster within a specific field was recognized by identifying the presence of overlapping 10–100 pixels.

## Western blot analysis

Saline and thioperamide-treated heterozygous mutant Hdh$^{Q7/Q111}$ and WT Hdh$^{Q7/Q7}$, mice were killed by cervical dislocation at 6 months of age, after behavioral assessment. Brains were quickly removed, dissected, frozen in dry ice and stored at −80°C until use. Protein extraction (n = 5–9 per group, only males) and western blot analysis were performed as previously described (*Puigdellívol et al., 2015*). The primary antibody 1C2 (1:1,000, Millipore) was used. Loading control was performed by reproving the membranes with an antibody to α-actin (1:20,000, MP Biochemicals). ImageJ software was used to quantify the different immunoreactive bands relative to the intensity of the α- actin band in the same membranes within a linear range of detection for the enhanced chemiluminescent kit reagent. Data are expressed as the mean ± SEM of band density.

## Statistical analysis

All the results were analyzed using GraphPad Prism software version 6.0. Data were presented as mean ± standard error of the means (SEM). Statistical analysis was performed using the unpaired two-sided Student's *t* test (95% confidence), one-way ANOVA or two-way ANOVA with the Bonferroni's *post hoc* test. Values of p<0.05 were considered statistically significant.

## Acknowledgements

We are very grateful to Ana Lopez (María de Maeztu Unit of Excellence, Institute of Neurosciences, University of Barcelona, MDM-2017–0729, Ministry of Science, Innovation and Universities) for mouse colony assistance. Dr. Teresa Rodrigo and the staff of the animal care facility (Facultat de Psicologia, Universitat de Barcelona), for their support and advice. We thank Manel Bosch at UB and Paul Thomas at the Henry Welcome Laboratory for Cell Imaging at UEA for their help with the microscopy. This work was supported by grants from Ministerio de Economia y Competitividad (RTI2018-094374-B-I00 to SG, SAF2017-88076-R to JA, RTI2018-095311-B-I00 to MG, and SAF-2017–87629 R to EC) Centro de Investigacion Biomédicas en Red sobre Enfermedades Neurodegenerativas (CIBERNED); RETICS (Red de Terapia Celular; RD16/0011/0012); Grant 20140610 from Fundació La Marató de TV3 to EC; RSC Grant Project RG140118; Jerome LeJeune Foundation FJL-01/01/2013; BBSRC BB/N504282/3 and start-up funds from QMUL.

## Additional information

### Funding

| Funder | Grant reference number | Author |
| --- | --- | --- |
| MRC | MR/S022465/1 | Peter J McCormick |
| RSC Grant Project | RG140118 | Peter J McCormick |
| BBSRC | BB/N504282/3 | Peter J McCormick |

| | | |
|---|---|---|
| Ministerio de Economía y Competitividad | RTI2018-094374-B-I00 | Silvia Ginés |
| Fundació la Marató de TV3 | 20140610 | Enric I Canela |
| Jerome Lejeune Foundation | FJL-01/01/2013 | Peter J McCormick |
| Ministerio de Economía y Competitividad | SAF2017-88076-R | Jordi Alberch |
| Ministerio de Economía y Competitividad | RTI2018-095311-B-I00 | Manuel Guzmán |
| National Institute on Drug Abuse | Supported by the intramural funds of the National Institute on Drug Abuse | Sergi Ferré |

The funders had no role in study design, data collection and interpretation, or the decision to submit the work for publication.

## Author contributions

David Moreno-Delgado, Conceptualization, Formal analysis, Investigation, Methodology, Performed and analyzed viability, calcium, internalization and organotypic culture experiments, Designed the experiments, analysed the results and wrote the manuscript; Mar Puigdellívol, Conceptualization, Formal analysis, Investigation, Methodology, Data curation, Performed all the treatments in mice, Conducted and analyzed the behavioral tests, Obtained all the tissue samples and prepared tissue slices for PLA, organotypic culture and mRNA experiments, Performed and analyzed western blot experiments and conducted and analysed spinophilin-immureactive experiments, Designed the experiments, analysed the results and wrote the manuscript.; Estefanía Moreno, Investigation, Performed PLA experiments and PLA quantification; Mar Rodríguez-Ruiz, Investigation, Assisted with function and viability experiments in cells and organotypic culture; Joaquín Botta, Investigation, Performed the binding experiments and assisted with calcium and cell death experiments; Paola Gasperini, Investigation, Performed all the shRNA related experiments and conducted and analyzed mRNA experiments; Anna Chiarlone, Investigation, Helped with the R6 and human PLA experiments; Lesley A Howell, Resources, Designed, synthesized and purified the disrupting peptides; Marco Scarselli, Resources, Aided with the cell experiments; Vicent Casadó, Antoni Cortés, Investigation, Performed and analyzed binding experiments; Sergi Ferré, Resources, Aided with the disrupting peptide experiments; Manuel Guzmán, Resources, Funding acquisition, Provided R6 mice samples and all the human samples, Discussed the results and edited the manuscript; Carmen Lluís, Supervision, Designed, supervised experiments, discussed, and helped write the manuscript; Jordi Alberch, Funding acquisition, Aided with the in vivo experiments; Enric I Canela, Resources, Funding acquisition, Aided with the disrupting peptide experiments; Silvia Ginés, Peter J McCormick, Conceptualization, Formal analysis, Supervision, Funding acquisition, Project administration, Conceived the idea, designed, supervised and coordinated the project, analyzed the results, and wrote the manuscript

## Author ORCIDs

Mar Puigdellívol https://orcid.org/0000-0002-9955-3558
Joaquín Botta https://orcid.org/0000-0001-6450-1267
Jordi Alberch https://orcid.org/0000-0002-8684-2721
Enric I Canela https://orcid.org/0000-0003-4992-7440
Silvia Ginés https://orcid.org/0000-0002-9479-8185
Peter J McCormick https://orcid.org/0000-0002-2225-5181

## Ethics

Animal experimentation: All procedures involving animals were performed in compliance with the National Institutes of Health Guide for the Care and Use of Laboratory Animals, and approved by the local animal care committee of the Universitat de Barcelona (99/01) and Generalitat de Catalunya (99/1094), in accordance with the European (2010/63/EU) and Spanish (RD53/2013) regulations for the care and use of laboratory animals. All protocols involving postmortem human sample were approved by the institutional ethic committees.

Decision letter and Author response
Decision letter https://doi.org/10.7554/eLife.51093.sa1
Author response https://doi.org/10.7554/eLife.51093.sa2

## Additional files

### Supplementary files

• Supplementary file 1. $H_3R$ and $D_1R$ binding parameters in STHdh$^{Q7}$, STHdh$^{Q111}$ cells. Striatal cells were homogenized in 50 mM Tris-HCl buffer and ligand binding was performed with membrane suspension (see materials and methods). Binding parameters from saturation and competition curves were obtained using Grafit software by fitting the binding data to the equation previously deduced (equation (3) in *Gracia et al., 2013*. Data are mean ± SEM of experiments performed per triplicate.

• Supplementary file 2. $H_3R$ and $D_1R$ binding parameters in different brain regions from 8-month-old Hdh$^{Q7/Q7}$ and Hdh$^{Q7/Q111}$ mice. Mouse striatal, cortical or hippocampal tissue were homogenized in 50 mM Tris-HCl buffer and ligand binding was performed with membrane suspension (see online methods). Binding parameters from saturation and competition curves were obtained using Grafit software by fitting the binding data to the equation previously deduced (equation (3) in *Gracia et al., 2013*. Data are mean ± SEM of experiments performed per triplicate (n = 6 Hdh$^{Q7/Q7}$ and n = 5 Hdh$^{Q7/Q111}$).

• Supplementary file 3. $H_3R$ and $D_1R$ mRNA expression levels the striatum of 4- and 8-month-old Hdh$^{Q7/Q7}$ and Hdh$^{Q7/Q111}$ mice. RT-PCR was performed in striatal extracts from Hdh$^{Q7/Q7}$ and Hdh$^{Q7/Q111}$ at 4 and 8 months of age as described in materials and methods. Results were normalized to actin gene expression. Data represent mean ± SEM (n = 3–4) of experiments performed in duplicate and are expressed as fold change of wild-type animals. Student's two-tailed *t* test was performed.

• Transparent reporting form

### Data availability

All data generated or analysed during this study are included in the manuscript and supporting files.

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
