## [Decision Letter]

**Acceptance summary:**

In this paper the authors identify a hetero-dimer consisting of the dopamine D_1_ and the histamine H_3_ receptor as a crucial component in early stages of Huntington's disease. They show that inhibiting the histamine H_3_ receptor prevents cell death signalling in cell culture and cognitive deficits in a mouse model. These data open a new avenue towards targeting Huntington's disease in its early stages.

**Decision letter after peer review:**

Thank you for submitting your article "Modulation of dopamine D_1_ receptors via histamine H_3_ receptors is a novel therapeutic target for Huntington's disease" for consideration by *eLife*. Your article has been reviewed by three peer reviewers, including Volker Dötsch as the Reviewing Editor and Reviewer #1, and the evaluation has been overseen by Richard Aldrich as the Senior Editor. The following individual involved in review of your submission has agreed to reveal their identity: Aylin C Hanyaloglu (Reviewer #3).

The reviewers have discussed the reviews with one another and the Reviewing Editor has drafted this decision to help you prepare a revised submission.

Summary:

This study by Moreno-Delgado and colleagues has identified a highly novel therapeutic target for Huntington's disease, a condition where there is even currently no drug to delay, let alone reverse this genetic condition. GPCR heteromers are an under-exploited therapeutic target, in part because of studies to demonstrate their direct in vivo functions, and their role in human disease have been limited. This study presents an incredible array of approaches to rigorously provide such evidence from in vitro, to in vivo HD animal models, including the existence of these heteromers in post-mortem patient samples. A critical and unique finding of the authors is that timing is critical for a therapeutic window in targeting these complexes.

Essential revisions:

1) One point that was discussed among the reviewers is the mechanism of heterodimer regulation by internalization and the distribution of the heterodimers. The distribution seen in the PLA assays show spots of the heterodimer instead of a wide distribution as one would expect from a protein in the plasma membrane. It is not clear if this distribution detects internalized receptors. Can internalization in cell culture be inhibited to show that the PLA assays detect heterodimers on the plasma membrane? Or, can the D_1_R-H_3_R heteromers expressed in STHdh^Q7^ and STHdh^Q111^ cells be detected with non-permeabilized condition (Figure 1)? In Figure 3, is anything known about the mechanisms underlying the decoupling of D1-H3 heterodimers in Hdh^Q7/Q111^ mutant mice?

2) It is suggested by the authors that activation of D_1_R with high concentrations of ligand induces internalization and disrupts the heteromer interaction with H_3_R. This may be the case as the paper cited Kotowski et al., 2011, do suggest that only very high ligand concentrations would induce receptor internalization to activate G protein signalling from an endosomal compartment, however, the data presented in Supplementary Figure 5A do not support that D_1_R undergoes internalization in their STHdh cells, as under basal conditions the receptor seems already vesicular/endosomal. This data that heteromer disruption is due to D_1_R internalization should be removed from Results/Discussion and should rather be discussed as a potential mechanism in the Discussion section.

3) The authors include nice controls in their in vitro systems for the H_3_R antibody. They should provide the same for the D_1_R antibody, or a reference if performed by the group in previous studies.

4) If higher concentrations of thioperamide are necessary for the crosstalk, please discuss the potential limitation of such strategy in therapeutic purpose.

5) In the second paragraph of the Discussion, it is suggested that more than one H_3_R ligand was used in the study, however, only data with thioperamide are presented. There is a mention, in the fourth paragraph of the Discussion, of an alternate H_3_R antagonist, VUF5681, and in the figure legend of Figure 4—figure supplement 1, but no data with VUF5681 in Figure 4—figure supplement 1. If the authors have these data it should be included to support selectivity of the thioperamide data to H_3_R. Similarly, thioperamide is also an antagonist of the H_4_ receptor. Can the authors exclude effects of inhibiting the H_4_ receptor?

---

## [Author Response]

Essential revisions:1) One point that was discussed among the reviewers is the mechanism of heterodimer regulation by internalization and the distribution of the heterodimers. The distribution seen in the PLA assays show spots of the heterodimer instead of a wide distribution as one would expect from a protein in the plasma membrane. It is not clear if this distribution detects internalized receptors. Can internalization in cell culture be inhibited to show that the PLA assays detect heterodimers on the plasma membrane? Or, can the D_1_R-H_3_R heteromers expressed in STHdh^Q7^ and STHdh^Q111^ cells be detected with non-permeabilized condition (Figure 1)? In Figure 3, is anything known about the mechanisms underlying the decoupling of D1-H3 heterodimers in Hdh^Q7/Q111^ mutant mice?

The cellular distribution pattern of PLAs is a problem that plagues this technique as judged by the literature (Borroto-Escuela et al. Mol Neurobiol. 2018; 55(8): 7038–7048. doi: 10.1007/s12035-018-0887-1 1 and Sebastianutto, I. et al. J Clin Invest 130, (2020) doi: 10.1172/JCI126361.) and in our own experience (Vinals et al., 2015 and Moreno et al. Neuropsychopharmacology 2017). In general, and due to the way the fluorescent signal is produced, PLA, are always puncta and so it is hard to judge if complexes are membrane or internal using this technique. We have only used the PLA as a tool, similar to co-immunoprecipitation (sans detergent) to demonstrate that these receptors can form a complex in vitro and ex vivo. Unfortunately, the epitopes of these antibodies are both intracellular so, permeabilization is required for these antibodies to work.

Currently, we do not understand what regulates D1-H3 heterodimers and why they would be lost in the Hdh^Q7/Q111^ mice. Now that we have made this discovery and understood that timing is crucial we would like to follow this up in future work.

2) It is suggested by the authors that activation of D_1_R with high concentrations of ligand induces internalization and disrupts the heteromer interaction with H_3_R. This may be the case as the paper cited Kotowski et al., 2011, do suggest that only very high ligand concentrations would induce receptor internalization to activate G protein signalling from an endosomal compartment, however, the data presented in Supplementary Figure 5A do not support that D_1_R undergoes internalization in their STHdh cells, as under basal conditions the receptor seems already vesicular/endosomal. This data that heteromer disruption is due to D_1_R internalization should be removed from Results/Discussion and should rather be discussed as a potential mechanism in the Discussion section.

In light of the comments above and the comments here we have decided to remove any data about trafficking and simply discuss this as a potential mechanism in the Discussion section. We have pointed out potential mechanisms of what might control dimer formation, and mentioned trafficking as one option to be pursued (Discussion).

3) The authors include nice controls in their in vitro systems for the H_3_R antibody. They should provide the same for the D_1_R antibody, or a reference if performed by the group in previous studies.

The antibody used for D_1_R has been used by us and tested extensively in three previous publications (Rodríguez-Ruiz et al., 2016, Moreno et al., 2011 and 2014). In addition, it is from one of the most reputable GPCR antibody companies (Frontier Institute) where it was validated in knock-out animals.

4) If higher concentrations of thioperamide are necessary for the crosstalk, please discuss the potential limitation of such strategy in therapeutic purpose.

In this study we have focused on the target of D_1_R-H_3_R for treating Huntington’s disease and the fact that timing is crucial. We chose thioperamide for the in vivo experiments as there was a large body of literature at the time on the ADME of this compound that allowed for a more cost effective preliminary animal study to demonstrate the effectiveness of the target. Going forward, more effective compounds would need to be identified that are either more potent or more targeted to the D_1_R-H_3_R complex. Regardless, this study highlights that this complex offers and early intervention strategy to slow progression, something that might be attractive in conjunction with current gene therapy trials going forward.

5) In the second paragraph of the Discussion, it is suggested that more than one H_3_R ligand was used in the study, however, only data with thioperamide are presented. There is a mention, in the fourth paragraph of the Discussion, of an alternate H_3_R antagonist, VUF5681, and in the figure legend of Figure 4—figure supplement 1, but no data with VUF5681 in Figure 4—figure supplement 1. If the authors have these data it should be included to support selectivity of the thioperamide data to H_3_R.

We have included our data on VUF5681 showing that it also can revert D_1_ induced cell death in 5 months brain slices (new Figure 4—figure supplement 1). In addition, we have included a discussion on what we believe H_4_ is not involved for the observed effects. To summarize, in cells silenced of H_3_R or when D_1_R-H_3_R heteromers are lost in the mice (6-8 months of age). we lose any effects of thioperamide.

Similarly, thioperamide is also an antagonist of the H_4_ receptor. Can the authors exclude effects of inhibiting the H_4_ receptor?

While we cannot rule out the effects of inhibiting H_4_R by using thioperamide, it is important to notice that H_4_R expression is higher in peripheral tissues, whereas H_3_R is abundantly expressed in the brain (De Esch IJ, Thurmond RL, Jongejan A, Leurs R. The histamine H_4_ receptor as a new therapeutic target for inflammation. Trends Pharmacol Sci 26: 462–469, 2005., and Haas, Sergeeva and Selbach, 2008) and we have used brain slices to show the protection afforded by thioperamide treatment. Moreover, we found that thioperamide treatment prevented cognitive and motor learning deficits as well as the loss of heteromer expression in different brain areas of the HD mouse brain, supporting a majority role for H_3_R. Finally, in the experiments with 8 months old animals where there is no D1-H3 heteromer and there is no effect of thioperamide, we have every reason to believe H_4_R expression would remain and yet thioperamide has no effect.